# Kazrin promotes dynein/dynactin-dependent traffic from early to recycling endosomes

Ines Hernandez-Perez, Javier Rubio, Adrian Baumann, Henrique Girao[†], Miriam Ferrando, Elena Rebollo, Anna M Aragay*, María Isabel Geli*

Institute for Molecular Biology of Barcelona (IBMB, CSIC), Baldiri Reixac 15, Barcelona, Spain

**\*For correspondence:**
aarbmc@ibmb.csic.es (AMA);
mgfbmc@ibmb.csic.es
(MIsabelG)

**Present address:** [†]Coimbra
Institute for Clinical and
Biomedical Research (ICBR),
Faculty of Medicine, University of
Coimbra, Coimbra, Portugal

**Abstract** Kazrin is a protein widely expressed in vertebrates whose depletion causes a myriad of developmental defects, in part derived from altered cell adhesion and migration, as well as failure to undergo epidermal to mesenchymal transition. However, the primary molecular role of kazrin, which might contribute to all these functions, has not been elucidated yet. We previously identified one of its isoforms, kazrin C, as a protein that potently inhibits clathrin-mediated endocytosis when overexpressed. We now generated kazrin knock-out mouse embryonic fibroblasts to investigate its endocytic function. We found that kazrin depletion delays juxtanuclear enrichment of internalized material, indicating a role in endocytic traffic from early to recycling endosomes. Consistently, we found that the C-terminal domain of kazrin C, predicted to be an intrinsically disordered region, directly interacts with several early endosome (EE) components, and that kazrin depletion impairs retrograde motility of these organelles. Further, we noticed that the N-terminus of kazrin C shares homology with dynein/dynactin adaptors and that it directly interacts with the dynactin complex and the dynein light intermediate chain 1. Altogether, the data indicate that one of the primary kazrin functions is to facilitate endocytic recycling by promoting dynein/dynactin-dependent transport of EEs or EE-derived transport intermediates to the recycling endosomes.

## Editor's evaluation

In their paper, Hernandez-Perez et al. perform a detailed and solid analysis of kazrin, a widely expressed protein that appears to be involved in many diverse cellular processes, but whose exact function is unknown. The authors employ mouse embryonic fibroblasts and biochemistry to investigate the function of Kazrin and determine that Kazrin promotes the dynein/dynactin-dependent transport of early endosomes. These valuable findings will be of interest to those in the field of intracellular transport.

## Introduction

Kazrin is a highly conserved and broadly expressed vertebrate protein, which was first identified as a transcript present in the human brain (*Nagase et al., 1999*). The human kazrin gene is located on chromosome 1 (1p36.21) and encodes at least seven isoforms (A-F and K), generated by alternative splicing (*Groot et al., 2004*; *Nachat et al., 2009*; *Wang et al., 2009*). From those, kazrin C is the shorter isoform that constitutes the core of all other versions, which bear N or C-terminal extensions. Since its discovery, several laboratories have reported a broad range of roles for the different kazrin isoforms in a myriad of experimental model systems. Thus, in humans, kazrin participates in structuring the skin-cornified envelope and it promotes keratinocyte terminal differentiation (*Groot et al., 2004*;

*Sevilla et al., 2008a*). In U373MG human astrocytoma cells, kazrin depletion leads to caspase activation and apoptosis (*Wang et al., 2009*). In *Xenopus* embryos instead, kazrin depletion causes ectoderm blisters (*Sevilla et al., 2008b*), as well as craniofacial development defects (*Cho et al., 2011*), linked to altered cell adhesion (*Sevilla et al., 2008b*), impaired epidermal to mesenchymal transition (EMT) and defective migration of neural crest cells (*Cho et al., 2011*). The subcellular localization of kazrin recapitulates its functional diversity. Depending on the isoform and cell type under analysis, kazrin associates with desmosomes (*Groot et al., 2004*), *adherens* junction components (*Cho et al., 2010*; *Sevilla et al., 2008a*), the nucleus (*Groot et al., 2004*; *Sevilla et al., 2008a*), or the microtubule cytoskeleton (*Nachat et al., 2009*). At the molecular level, the N-terminus of kazrin, predicted to form a coiled-coil, directly interacts with several p120-catenin family members (*Sevilla et al., 2008b*), as well as with the desmosomal component periplakin (*Groot et al., 2004*), and it directly or indirectly regulates RhoA activity (*Groot et al., 2004*; *Sevilla et al., 2008a*; *Cho et al., 2010*). How kazrin orchestrates such many cellular functions at the molecular level is far from being understood.

We previously identified human kazrin C as a protein that potently inhibits clathrin-mediated endocytosis when overexpressed (*Schmelzl and Geli, 2002*). In the present work, we generated kazrin knock out (kazKO) Mouse Embryonic Fibroblasts (MEFs) to analyze its role in endocytic traffic in detail. We found that depletion of kazrin caused accumulation of peripheral EEs and delayed transfer of endocytosed transferrin (Tfn) to the pericentriolar juxtanuclear region, where the recycling endosomes (REs) usually concentrate (*Granger et al., 2014*; *Tang and Marshall, 2012*). Consequently, cellular functions requiring intact endosomal traffic through the REs, such as cell migration and cytokinetic abscission, were also altered in kazKO cells. Consistent with its role in endocytic traffic, we found that the kazrin C C-terminal portion predicted to be an intrinsically disordered region (IDR), interacted with different components of the EEs, it was required to form *foci* on these organelles and it was necessary to sustain efficient transport of internalized Tfn to the juxtanuclear region. Further, the N-terminus of kazrin C shared considerable homology with dynein/dynactin activating adaptors, and kazrin directly interacted with the dynactin complex and the dynein light intermediate 1 (LIC1). The data thus suggested that kazrin facilitates the transfer of endocytosed material to the pericentriolar REs by promoting retrograde dynein/dynactin-dependent transport of EEs or EE-derived transport intermediates.

## Results

### Kazrin depletion impairs endosomal traffic

We originally identified kazrin C as a human brain cDNA, whose overexpression causes accumulation of the transferrin receptor (TfnR) at the plasma membrane (PM) in Cos7 cells (*Schmelzl and Geli, 2002*), suggesting that kazrin might be involved in clathrin-mediated endocytic uptake from the PM. However, treatment of Cos7 cells with an shRNA directed against kazrin (sh*Kzrn*) (*Figure 1—figure supplement 1A*) did not inhibit endocytic internalization but it rather increased the intracellular signal of Alexa 647-Tfn (A647-Tfn) upon a 2 hr incubation (*Figure 1—figure supplement 1B,C*), indicating that depletion of kazrin either exacerbated endocytic uptake or inhibited endocytic recycling. The distribution of A647-Tfn labeled endosomes was also altered in the sh*Kzrn* treated cells, as compared with that of untreated cells or cells transfected with a control shRNA (shCTR). In wild-type (WT) and shCTR-treated cells, A647-Tfn accumulated in the juxtanuclear region, where the RE is usually located (*Granger et al., 2014*; *Sheff et al., 2002*; *Shen et al., 2006*; *Tang and Marshall, 2012*; *Yamashiro et al., 1984*). In contrast, TxR-Tfn labeled endosomes appeared more scattered toward the cell periphery in sh*Kzrn* treated cells (*Figure 1—figure supplement 1B,C*). The accumulation of endocytosed material at the periphery suggested that kazrin plays a post-internalization role in the endocytic pathway, possibly in the transport of material toward the juxtanuclear RE.

shRNA transfection in Cos7 cells did not achieve complete kazrin depletion in a reproducible manner and it hampered complementation. To overcome these problems, we generated kazrin knockout MEFs (kazKO MEFs) using the CRISPR-Cas9 technology and we used a lentiviral system to subsequently create two cell lines that expressed GFP or GFP-kazrin C upon doxycycline induction (*Figure 1—figure supplement 2A*). Immunoblot analysis demonstrated that the expression level of GFP-kazrin C in the absence of doxycycline or upon a short overnight (up to 12 hr) incubation was similar to that of the endogenous kazrin (low expression, 1–4 times the endogenous kazrin expression

level) (*Figure 1—figure supplement 2B*). Under these conditions, the GFP-kazrin C was barely detectable by fluorescence microscopy. This might explain why none of the commercially available or home-made anti-kazrin antibodies detected a specific signal in WT MEFs. Doxycycline incubation for longer periods (up to 24 hr induction) resulted in moderate expression (4–8 times the endogenous kazrin expression levels) (*Figure 1—figure supplement 2B*), but allowed us to clearly visualize its localization by microscopy (*Figure 1—figure supplement 2C*).

To better discern on the possible effects of kazrin depletion on endocytic uptake or in subsequent trafficking events, WT and kazKO cells were exposed to a short, 10 min incubation pulse with Texas Red-Tfn (TxR-Tfn), fixed, and analyzed. In WT cells, TxRed-Tfn accumulated in a pericentriolar region adjacent to the nucleus, similar to Cos7 cells (*Figure 1A* and *Figure 1—figure supplement 3A*). No differences in the amount of internalized TxR-Tfn were observed between WT and kazKO MEF (*Figure 1—figure supplement 4*), suggesting that kazrin did not play a relevant role in the formation of endocytic vesicles from the PM, but it might rather work downstream in the pathway. In agreement with this view, and similar to the sh*Kzrn* Cos7, kazKO MEFs accumulated TxR-Tfn in the cell periphery, as compared with WT cells (*Figure 1A and B*). Juxtanuclear accumulation of TxR-Tfn was restored in kazKO MEF by low, physiological expression of GFP-kazrin C but not GFP (*Figure 1A and B*), indicating a direct role of kazrin in the process. No significant difference between the kazKO and the kazKO GFP-expressing cells could be detected in these experiments. Therefore, in order to simplify the experimental design, further assays were normalized to the closest isogenic kazKO background, namely the kazKO cells when compared to the WT, and the kazKO GFP expressing cells when compared to kazKO MEF expressing GFP-kazrin C.

To evaluate if the scattering of TxR-Tfn endosomes was due to a defect in the transfer of material form the EEs to the REs or if it was caused by the dispersal of the REs, we analyzed the distribution of the EE and the RE markers EEA1 (Early endosome autoantigen 1) and RAB11 (Ras-related in brain 11), respectively. We observed that kazKO MEFs accumulated peripheral, often enlarged, EEA1 positive structures, as compared with WT MEF (*Figure 1C and D* and *Figure 1—figure supplement 3B*). The juxtanuclear distribution of the REs, was however not significantly affected in the knock-out cells (*Figure 1—figure supplement 5*). Again, low expression of GFP-kazrin C but not GFP recovered the EEA1 juxtanuclear distribution (*Figure 1C and D*). The data thus suggested that kazrin promotes the transfer of endocytosed material toward the juxtanuclear region, where the RE is located.

Consistent with the role of kazrin in endocytic traffic towards the RE, recycling of TxR-Tfn back to the PM was diminished in kazKO cells (*Figure 1E*), albeit not completely blocked. A complete block in recycling was not to be expected because, in addition to the RAB11 route, the TfnR can take a RAB4-dependent shortcut to the PM (*Sheff et al., 2002*). As for the juxtanuclear Tfn enrichment assays, the expression of GFP-kazrin C but not GFP restored the recycling defects installed in the kazKO MEF (*Figure 1E*).

To further confirm the specific role of kazrin in endocytic recycling via the juxtanuclear RE, we analyzed its implication in cellular processes that strongly rely on this pathway, such as cell migration and invasion (*Emery and Ramel, 2013*; *Fan et al., 2004*; *Jones et al., 2006*; *Kessler et al., 2012*; *Mammoto et al., 1999*; *Powelka et al., 2004*; *Ramel et al., 2013*; *Wilson et al., 2018*; *Yoon et al., 2005*). Analysis of the migration of single WT and kazKO cells through Matrigel demonstrated that depletion of kazrin significantly reduced the migration speed, which, similar to endocytic traffic, was recovered upon re-expressing GFP-kazrin C at low levels, but not GFP (*Figure 2A and B*, *Video 1*). We also observed an increased persistency in the migration of kazKO cells (*Figure 2—figure supplement 1*), but it was not recovered with GFP-kazrin C re-expression (*Figure 2—figure supplement 1*). Increased persistency might be a secondary effect caused by the trafficking block to the RE, which accelerates recycling via the RAB4-dependent shortcut circuit (*Perrin et al., 2013*; *White et al., 2007*). The long recycling pathway also plays an important role in the last abscission step during cytokinesis (*Fielding et al., 2005*; *Horgan et al., 2004*; *Pollard and O'Shaughnessy, 2019*; *Wilson et al., 2005*). Consistent with kazrin playing a role in this process, kazKO cells had a significant delay in cell separation after cytokinesis, which was again restored by GFP-kazrin C expression (*Figure 2C and D* and *Video 2*).

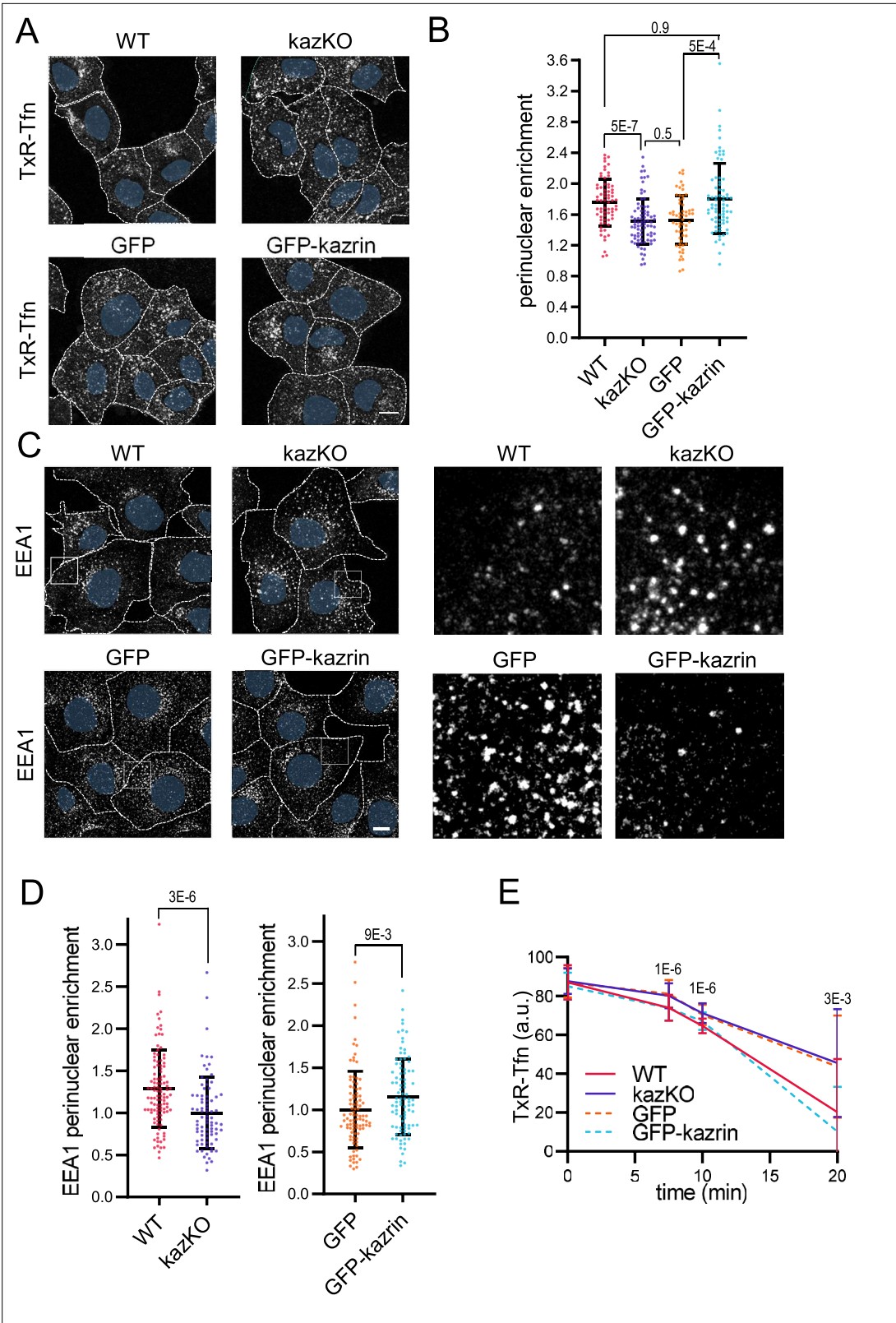

**Figure 1.** Kazrin depletion impairs endosomal traffic. (**A**) Confocal images of wild-type (WT) and kazKO MEF or kazKO MEF expressing low levels (See Materials and methods (M & M)) of GFP or GFP-kazrin C, incubated with Texas Red-Tfn (TxR-Tfn) at 16 °C and chased at 37 °C for 10 min. Scale bar, 10 μm. Cell borders are indicated by dashed lines and nuclei in blue. (**B**) Scattered plot of the mean ± SD (Standard deviation) TxR-Tfn perinuclear enrichment (See M & M) for the cells described in A, after 10 min incubation at 37 °C. *p*-values of the two-tailed Mann-Whitney tests are shown.

*Figure 1 continued on next page*

*Figure 1 continued*

n>58 cells for each sample. Refer also to *Figure 1—figure supplement 1* for the effects of kazrin depletion in Cos7 cells, *Figure 1—figure supplement 2* for the strategy of kazKO MEF generation, *Figure 1—figure supplement 3A* for pericentriolar localization of internalized transferrin (Tfn) in WT cells, and *Figure 1—figure supplement 4* for the effects of kazrin depletion on TxR-Tfn uptake in MEF. (**C**) Confocal images of the WT and kazKO MEF, or kazKO MEF expressing low levels of GFP or GFP-kazrin C, fixed and stained with anti-EEA1 and A568-conjugated secondary antibodies. A 17 µm$^2$ magnified insets showing endosomes in the peripheral areas are shown on the right. Scale bar, 10 µm. Cell borders are indicated with dashed lines and nuclei in blue. (**D**) Scattered plots of the mean ± SD early endosome autoantigen 1 (EEA1) perinuclear enrichment (See M & M) in the cells described in C. The values were normalized to the corresponding kazKO cells (either kazKO or kazKO GFP). *p*-values of the two-tailed Mann-Whitney tests are shown. n>80 cells for each sample. Refer to *Figure 1—figure supplement 3B* for pericentriolar localization of EEA1 in WT cells and *Figure 1—figure supplement 4* for the effects of kazrin depletion on the RAB11 perinuclear enrichment. (**E**) Line plot of the mean ± SD TxR-Tfn fluorescence intensity per cell in WT and kazKO MEFs, or kazKO MEFs expressing low levels of GFP and GFP-kazrin C, at the indicated time points after loading early endosomes (EEs) with TxR-Tfn at 16 °C and release at 37 °C to allow recycling (See M & M for further details). Data were normalized to the average intensity at time 0. *p*-values of the two-tailed Student *t*-tests are shown. n>16 cells per sample and time point.

The online version of this article includes the following source data and figure supplement(s) for figure 1:

**Source data 1.** Data for graphs presented in *Figure 1B, D and E*.

**Figure supplement 1.** Kazrin depletion in Cos7 cells alters endosomal trafficking.

**Figure supplement 1—source data 1.** Un-cropped blots for *Figure 1—figure supplement 1A*.

**Figure supplement 1—source data 2.** Data for graphs presented in *Figure 1—figure supplement 1C*.

**Figure supplement 2.** Generation of kazKO MEF and kazKO MEF expressing GFP or GFP-kazrin C.

**Figure supplement 2—source data 1.** Un-cropped blots for *Figure 1—figure supplement 2B*.

**Figure supplement 3.** Texas Red-Tfn (TxR-Tfn) and early Endosomes endosomes (EEs) concentrate in a pericentriolar juxtanuclear region in WT MEF.

**Figure supplement 4.** Depletion of kazrin does not impair Texas Red-Tfn (TxR-Tfn) uptake.

**Figure supplement 4—source data 1.** Data for graph presented in *Figure 1—figure supplement 4B*.

**Figure supplement 5.** Depletion of kazrin does not significantly alter the distribution of Ras-Related in Brain 11 (RAB11) compartments.

**Figure supplement 5—source data 1.** Data for graph presented in *Figure 1—figure supplement 5B*.

## Kazrin is recruited to EEs and directly interacts with components of the endosomal machinery through its C-terminal predicted IDR

Next, we investigated whether endogenous kazrin was present in EEs. For that purpose, we initially used subcellular fractionation and immunoblot because the endogenous protein was not detectable by fluorescence microscopy, nor was GFP-kazrin C expressed at physiological levels. As shown in *Figure 3A*, endogenous kazrin neatly co-fractionated in the lightest fractions with EE markers such as the tethering factor EEA1 and EHD (Eps15 homology domain) proteins, most likely corresponding to EHD1 and EHD3. On the contrary, it only partially co-fractionated with a transitional early-to-late endosome marker (Vacuolar Protein Sorting 35 ortholog, VPS35) and did not with markers of recycling endosomes (RAB11) or the Golgi apparatus (Golgi Matrix protein 130, GM130) (*Figure 3A*). Moderately overexpressed GFP-kazrin C also co-fractionated with EEs, although it appeared slightly more spread towards the RE and Golgi fractions in the gradient (*Figure 3A*). Endogenous kazrin localization at EEs was confirmed by subcellular fractionation experiments in mIMCD3 cells (*Figure 3—figure supplement 1*).

To confirm the interaction of kazrin C with endosomes, we immunoprecipitated GFP-kazrin C from native cellular extracts and probed the immunoprecipitates for a number of proteins involved in endosomal trafficking. We detected specific interactions of GFP-kazrin C and γ-adaptin, a component of the Golgi and endosomal clathrin adaptor AP-1 (Adaptor Protein 1), as well as clathrin and EHD proteins (*Figure 3B*). No interaction with the retromer subunit VPS35, the tethering factor EEA1, or the clathrin adaptors GGA2 (Golgi-localized Gamma-ear-containing ADP-ribosylation factor-binding 2) or AP-2 (Adaptor Protein 2) could be detected in immunoprecipitation assays (*Figure 3B*), indicating that kazrin C binds the machinery implicated in endosomal traffic from EEs to or through REs (*Caplan et al., 2002*; *George et al., 2007*; *Grant et al., 2001*; *Grant and Caplan, 2008*; *Jović et al., 2007*; *Lin et al., 2001*; *Naslavsky et al., 2006*; *Perrin et al., 2013*; *Rapaport et al., 2006*). Pull-down assays with purified components demonstrated that kazrin C can directly interact with EHD1 and EHD3, the clathrin heavy chain terminal domain, and the γ-adaptin ear (*Figure 3C*). Pull-down assays from cell extracts showed that the EHD proteins and the AP-1 complex bound to the

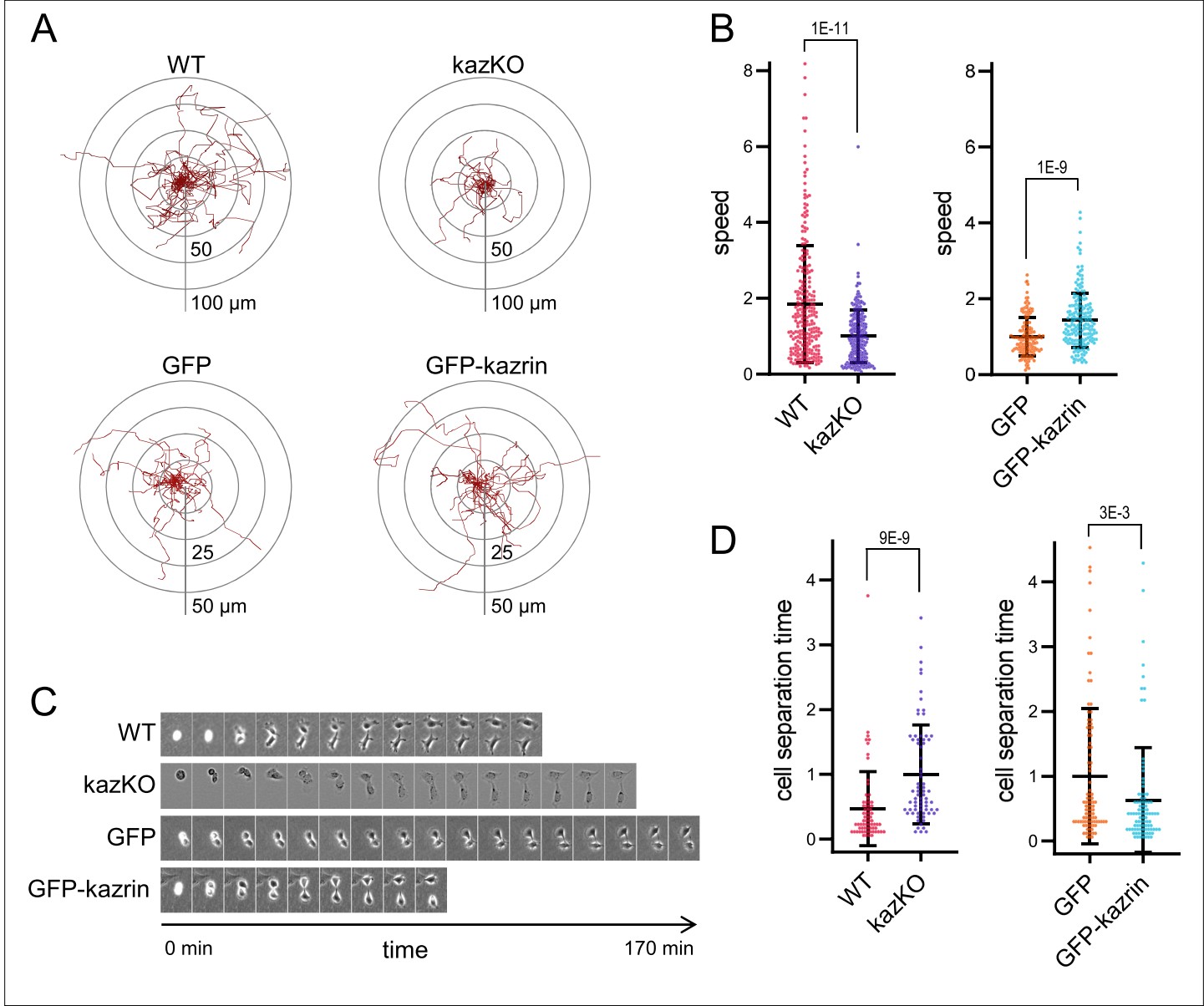

**Figure 2.** Kazrin depletion impairs cell migration and division. (**A**) Paths described by individually migrating wild-type (WT) and kazKO MEF or kazKO MEFs expressing GFP or GFP-kazrin C at low levels (See M & M). The cells were embedded in Matrigel and tracked for 9 hr with a 10 min time lapse. All tracks start at the (0,0) coordinate in the graph. See also *Video 1* for examples of individual migrating cells. (**B**) Scattered plot of the mean ± SD speed of cells described in (**A**). The data wre normalized to the mean of the corresponding KO cells (either kazKO or kazKO expressing GFP). *p*-values of the two-tailed Mann-Whitney tests are shown. n>100 cells per condition. See also *Figure 2—figure supplement 1* for the effects of kazrin depletion on directionality. (**C**) Time-lapse epifluorescence images of WT and kazKO MEFs or kazKO MEFs expressing GFP or GFP-kazrin C at low levels, as they divide. Cells were recorded every 10 min. See also *Video 2* for examples of individual dividing cells. Windows are 55 x 74 μm$^2$ for WT MEF, 38 x 50 μm$^2$ for kazKO MEF and 60 x 80 μm$^2$ for GFP and GFP-kazrin C expressing MEF. (**D**) Mean time ± SD between substrate attachment and complete cell separation of the cells described in C. The data were normalized to the mean of the corresponding KO (kazKO or kazKO expressing GFP). *p*-values of the two-tailed Mann-Whitney tests are shown. n>68 dividing cells per condition.

The online version of this article includes the following source data and figure supplement(s) for figure 2:

**Source data 1.** Data for graphs presented in *Figure 2B and D*.

**Figure supplement 1.** Effect of kazrin depletion on the directionality ratio.

**Figure supplement 1—source data 1.** Data for graphs presented in *Figure 2—figure supplement 1*.

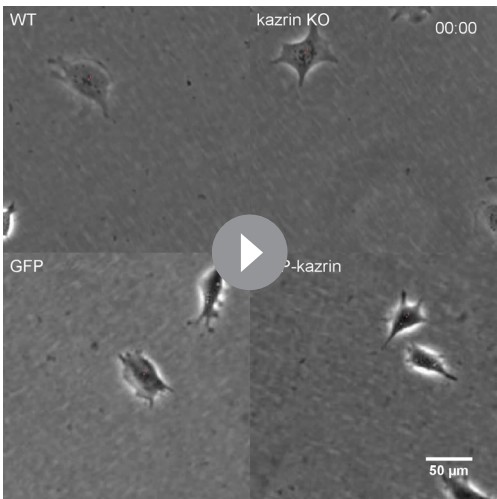

**Video 1.** Videos of individually migrating wild-type (WT) and kazKO MEF, and kazKO MEF expressing low levels of GFP and GFP-kazrin C. The cells were embedded in Matrigel and imaged with an epifluorescence microscope.

https://elifesciences.org/articles/83793/figures#video1

C-terminus of kazrin C, predicted to be an IDR, but not to the N-terminus (*Figure 3D and E*). Most kazrin interacting partners were previously defined to bind its N-terminal region predicted to form a coiled-coil (*Groot et al., 2004*; *Sevilla et al., 2008b*). The interaction of endogenous kazrin with γ-adaptin could also be confirmed in co-immunoprecipitation experiments from MEFs, using a polyclonal antibody against the C-terminus of kazrin C (*Figure 3—figure supplement 2*). In lipid overlay and liposome pelleting assays, we also found that purified kazrin C interacted with PhosphatidylInositol 3-Phosphate (PI3P) (*Figure 3F and G*), a lipid particularly enriched on EEs (*Gillooly et al., 2000*; *Wang et al., 2019*). The interaction required the poly-Lys stretch in the C-terminus of kazrin C (*Figure 3F*), previously proposed to constitute a nuclear localization signal (*Groot et al., 2004*). The data suggested that the predicted kazrin C IDR had multiple binding sites for EE components, and therefore, it might be required for its EE recruitment and its function in endosomal traffic.

To investigate the role of the C-terminal region of kazrin C in its recruitment to endosomes and its function in endocytic traffic, we generated kazKO cells expressing a GFP-kazrin C construct lacking the C-terminal predicted IDR (lacking amino acids 161–327) (kazKO GFP-kazrin C-Nt), using the lentivirus system (*Figure 4—figure supplement 1*). We then analyzed its subcellular localization and its capacity to complement the kazKO endocytic defects, as compared with full-length GFP-kazrin C or GFP. As shown in *Figure 4A and B*, moderately expressed GFP-kazrin C significantly associated with the microsomal fraction containing the EEs. In contrast, GFP and GFP-kazrin C-Nt were mostly cytosolic, indicating that the C-terminal predicted IDR, which binds PI3P, γ-adaptin, and EHD proteins, might be required to bring kazrin to cellular membranes. Next, we proceeded to image cells expressing moderate levels of GFP-kazrin C and GFP-kazrin C-Nt, upon loading of EEs with TxR-Tfn at 16 °C. The previously reported localizations of kazrin C in the nucleus and at cell-cell contacts were evident in these cells (*Figure 4—figure supplement 2*; *Groot et al., 2004*). At the PM, GFP-kazrin C neatly co-localized with the *adherens* junction components N-cadherin, β-catenin, and p120-catenin, but not with desmoglein, a desmosomal cadherin (*Figure 4—figure supplement 2*). In addition to the previously reported localizations, GFP-kazrin C formed small *foci*, which associated with the surface of the TxR-Tfn labeled endosomes (*Figure 4C and D*; *Figure 4—figure supplement 3* and *Video 3*). Co-localization of GFP-kazrin C *foci* with EHD-labeled structures could also be observed in the cell periphery (*Figure 4—figure supplement 4* and *Video 4*). GFP-kazrin C-Nt and GFP staining at similar expression levels appeared mostly cytosolic, with nearly no visible (for GFP) or scarce (for GFP-kazrin C-Nt) *foci* per cell (*Figure 4C to E*). The few GFP-kazrin C-Nt *foci* observable appeared less associated with TxR-Tfn loaded endosomes, as compared to GFP-kazrin C (*Figure 4C and D*; *Figure 4—figure supplement 3* and *Videos 3 and 5*).

The data thus indicated that the C-terminal predicted IDR was required to recruit kazrin C to endosomal membranes and consequently, it should be required to sustain its function in endosomal traffic, provided that it played a direct role in the process. To test this hypothesis, we

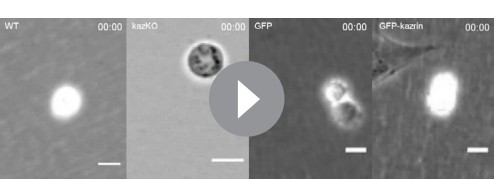

**Video 2.** Videos of dividing wild-type (WT) and kazKO MEF, and kazKO MEF expressing low levels of GFP and GFP-kazrin C, from the moment the mother cell attached to the substrate until the daughter cells were completely separated. Scale bar = 10 μm. The cells were embedded in Matrigel and imaged with an epifluorescence microscope.

https://elifesciences.org/articles/83793/figures#video2

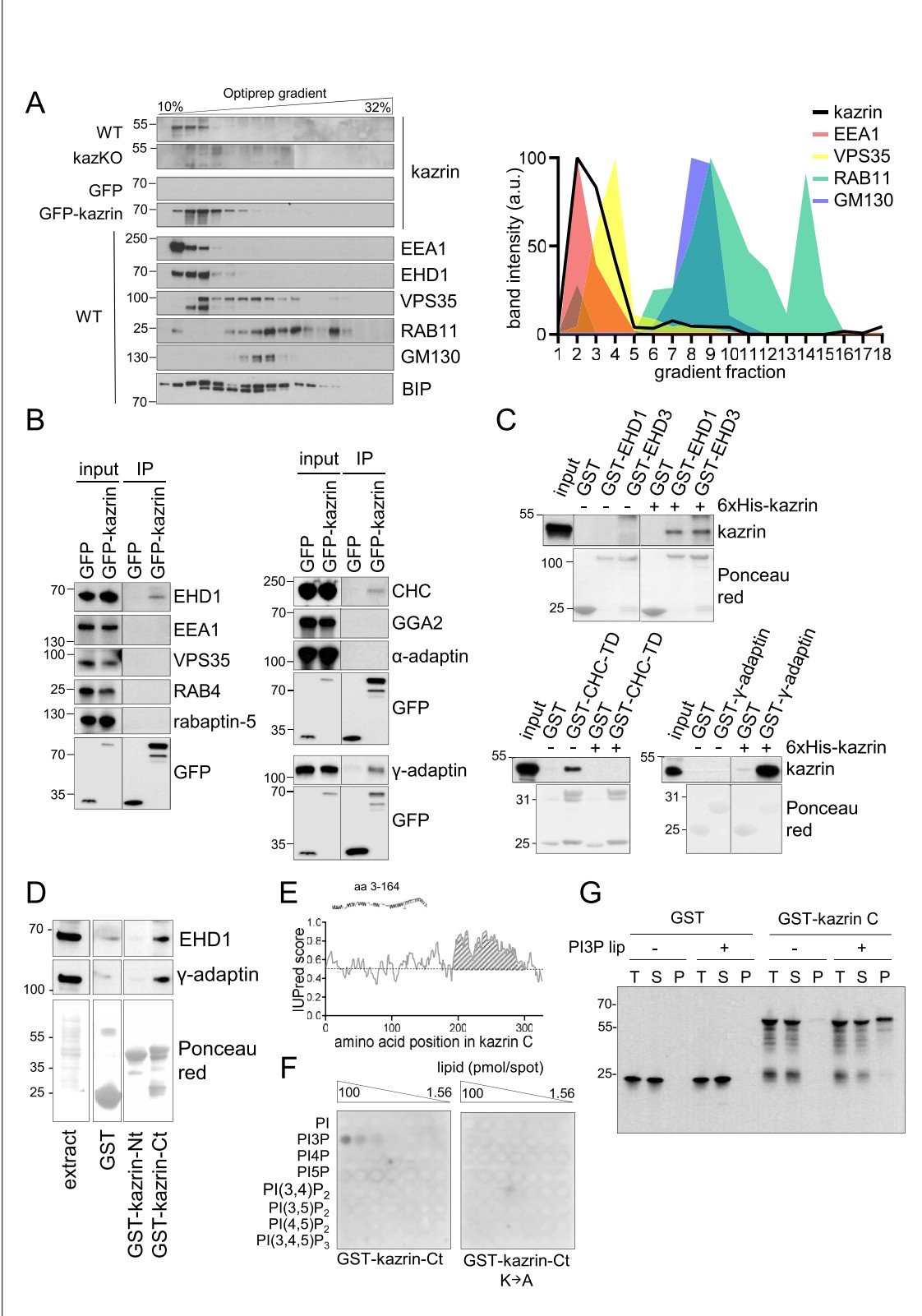

**Figure 3.** Kazrin is an endosomal protein. (**A**) Left, immunoblots of Optiprep density gradient fractionations of membrane lysates of wild-type (WT) and kazKO MEF or kazKO MEF moderately expressing (See M & M) GFP or GFP-kazrin C. The membranes were probed with antibodies against the kazrin C N-terminus, EEA1, and EHD1 (EE markers), VPS35 (RAB5/RAB7 transition endosomal marker), RAB11 (RE/Golgi marker), GM130 (*cis*-Golgi marker) and BIP (Binding immunoglobulin protein) (ER marker). The antibody against EHD1 is likely to recognize other Eps15 homology domain (EHD) proteins.

*Figure 3 continued on next page*

*Figure 3 continued*

Band intensity plots per fraction for kazrin or the indicated intracellular membrane markers are shown on the right. The signal intensities of each fraction were normalized to the maximum for each antibody. All gradients were loaded with the same amount of total protein. Refer also to *Figure 3—figure supplement 1* for co-fractionation of kazrin and early endosome autoantigen 1 (EEA1) in the lightest gradient fractions in IMCD3 cells. (**B**) Immunoblots of anti-GFP-agarose precipitates from lysates of kazKO MEF moderately expressing GFP or GFP-kazrin C, probed with antibodies against the indicated proteins. 10 µg of total protein were loaded as input. (**C**) Immunoblots of pull-downs from glutathione-Sepharose beads coated with GST, or GST fused to full-length EHD1 or EHD3, the clathrin heavy chain terminal domain (CHC-TD) or the γ-adaptin ear domain, incubated with purified 6xHis-kazrin C. The membranes were probed with an anti-kazrin antibody (ab74114, from Abcam) and stained with Ponceau red to visualize the GST fusion constructs. Refer also to *Figure 3—figure supplement 2* for evidence indicating co-immunoprecipitation of endogenous kazrin with γ-adaptin and clathrin. (**D**) Immunoblots of pull-downs from glutathione-Sepharose beads coated with GST, or GST fused to the N- (amino acids 1–174) or C- (amino acids 161–327) terminal portions of kazrin C, incubated with non-denaturing extracts from MEFs. 10 µg of total protein were loaded as input. Ponceau red staining of the same membrane (lower panels) is shown to visualize the protein extract or the GST fusion constructs. (**E**) Prediction of IDRs in kazrin C. The graph shows the probability of each residue of being part of an intrinsically disordered region (IDR), according to the IUPred2A software. Residues in the shaded area have a consistent probability over 0.5 to form part of an IDR. (**F**) Immunoblots of a lipid-binding assay performed with either the purified GST-kazrin C C-terminal portion (amino acids 161–327) (GST-kaz-Ct) or an equivalent construct in which the poly-K region has been mutated to poly-A. The membranes used in this assay contain a concentration gradient of the indicated phosphoinositides. Membranes were probed with an anti-GST antibody. (**G**) Immunoblot of a liposome pelleting assay probed with an anti-GST antibody. GST or GST-kazrin C were incubated in the presence (+) or absence (−) of liposomes containing 5% phosphatidylinositol 3-phosphate (PI3P). Liposomes were recovered at 100.000 g for 1.5 hr. One equivalent of the input (**T**), one equivalent of the supernatant (**S**), and ten equivalents of the pellet (**P**) were loaded per sample.

The online version of this article includes the following source data and figure supplement(s) for figure 3:

**Source data 1.** Un-cropped blots for *Figure 3A, B, C, D and G*.

**Figure supplement 1.** Endogenous kazrin co-fractionates with early endosome autoantigen 1 (EEA1) in the lightest fractions in IMCD3 cells.

**Figure supplement 1—source data 1.** Un-cropped blots for *Figure 3—figure supplement 1*.

**Figure supplement 2.** Antibodies raised against kazrin C co-immunoprecipitate γ-adaptin and clathrin.

**Figure supplement 2—source data 1.** Un-cropped blots for *Figure 3—figure supplement 2*.

---

investigated the capacity of GFP-kazrin C-Nt to restore traffic of TxR-Tfn to the juxtanuclear region in the kazKO cells, as compared to the full-length kazrin C. As shown in *Figure 4F and G*, GFP-kazrin C significantly increased the juxtanuclear enrichment of TxR-Tfn in a kazKO background upon a 10 min uptake, as compared to GFP expression, whereas GFP-kazrin C-Nt did not.

## Kazrin C localizes at the pericentriolar region and directly interacts with dynactin and LIC1

Interestingly, we observed that in most cells expressing GFP-kazrin C, one or two very bright *foci* embracing EEs were visible in the juxtanuclear region (*Figure 5A*). Neat co-localization of the bright juxtanuclear GFP-kazrin C *foci* with pericentrin demonstrated that GFP-kazrin C accumulated in the pericentriolar region (*Figure 5B*). Live-cell imaging evidenced small GFP-kazrin C *foci* moving towards and away from the pericentriolar region (*Figure 5—figure supplement 1* and *Videos 6 and 7*). These structures were reminiscent of pericentriolar satellites, which are IDR-enriched membrane-less compartments that transport centrosomal components in a microtubule-dependent manner (*Prosser and Pelletier, 2020*). Treatment with the microtubule depolymerizing drug nocodazole disrupted the juxtanuclear localization of GFP-kazrin C, along with that of EEs (*Figure 5C and D*). Likewise, treatment with ciliobrevin to inhibit dynein activity resulted in concomitant dispersal of EEs as previously observed (*Aniento et al., 1993*; *Burkhardt et al., 1997*; *Driskell et al., 2007*; *Flores-Rodriguez et al., 2011*) and GFP-kazrin C depletion from pericentrin condensates (*Figure 5E and F*).

Our observations suggested that dynein-dependent transport is required not only for the accumulation of EEs at the pericentriolar region as previously reported (*Aniento et al., 1993*; *Burkhardt et al., 1997*; *Driskell et al., 2007*; *Flores-Rodriguez et al., 2011*) but also, for the localization of GFP-kazrin C in this area. Interestingly, pericentriolar localization of GFP-kazrin C was reminiscent of that observed for well-established or candidate dynein/dynactin activating adaptors such as hook2, hook3, or BICDR1 (*Baron and Salisbury, 1988*; *Schlager et al., 2010*; *Szebenyi et al., 2007*). Indeed, kazrin C shared 23.3% identity and 57.3% similarity with BICDR1 (BICauDal Related protein 1) (*Figure 5G*), over 232 amino acids, and 19.6% identity and 61.3% similarity with hook3, over 168 amino acids. Such similarity was in the range of that shared between hook3 and BICDR1 (24.7% identity and 56.7% similarity over 268 amino acids) (LALIGN). Therefore, we hypothesized

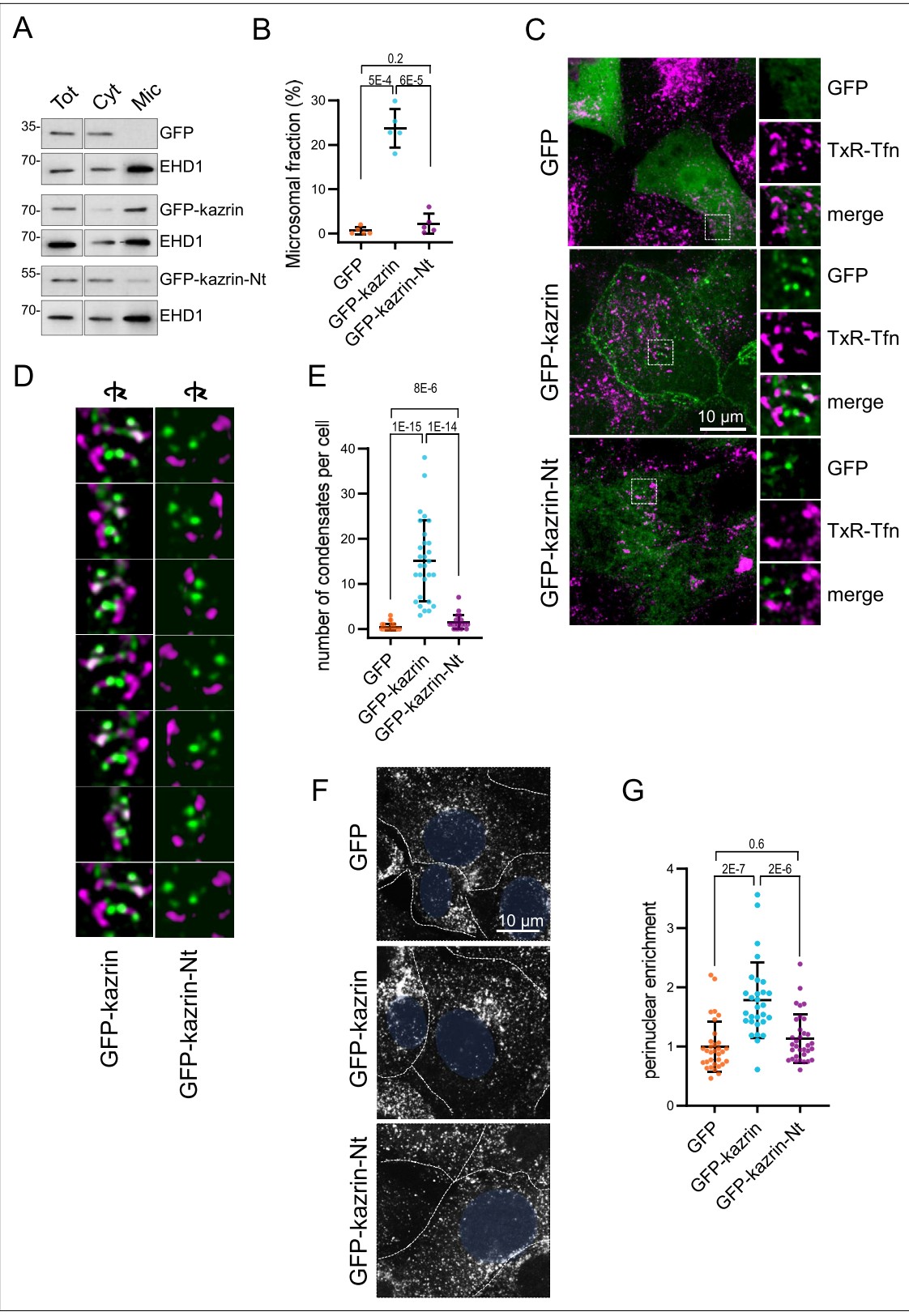

**Figure 4.** The predicted intrinsically disordered region (IDR) of kazrin C is required for its endocytic function. (**A**) Immunoblots of subcellular fractionations from kazKO cells expressing moderate amounts (See M & M) of GFP, GFP-kazrin C or a GFP-kazrin C construct lacking the C-terminal predicted IDR (GFP-kazrin C-Nt). Cells were lysed in a non-denaturing buffer and centrifuged at 186,000 g for 1 hr to separate membranes (Mic) from the cytosol (Cyt). 15 μg of the total extract (Tot), and 1 and 5 equivalents of the cytosolic and membrane fractions were loaded per lane, respectively.

*Figure 4 continued on next page*

*Figure 4 continued*

(**B**) Scattered plot of the mean ± SD percentage of the GFP-signal associated with the microsomal fraction (Mic) in kazKO MEF expressing moderate amounts of GFP, GFP-kazrin C or GFP-kazrin C-Nt. Student´s *t*-tests *p*-values are shown. n=5 independent experiments for each sample. See M & M for experimental details. (**C**) MIP of confocal images of kazKO MEF expressing moderate amounts of GFP, GFP-kazrin C, or GFP-kazrin-Nt, loaded with 20 µg/ml of Texas Red-Tfn (TxR-Tfn) at 16 °C to accumulate endocytic cargo on early Endosomes endosomes (EEs). The images from the GFP and TxR channels and the merge from 5 × 5 µm$^2$ fields are shown on the right. (**D**) Frames showing consecutive 60° turn snapshots of the 5 × 5 µm$^2$ 3D reconstruction animations of the insets shown in C for GFP-kazrin C and GFP-kazrin-C-Nt, showing the association of GFP-kazrin C *foci*, but not GFP-kazrin C-Nt, with TxR-Tfn-loaded EEs. (**E**) Scattered plot of the mean ± SD of the number of condensates per cell, visible with the GFP filter channel in the kazKO cells described in (**B**). *p*-values of the two-tailed Mann-Whitney test are shown. n=29 cells for each sample. Refer also to *Video 3* for four 3D reconstruction animations of TxR-Tfn loaded endosomes associated with GFP-kazrin C, and *Video 5* for GFP-kazrin C-Nt; *Figure 4—figure supplement 2* for co-localization of GFP-kazrin C with adhesion molecules; *Figure 4—figure supplement 3* for analysis of the association of GFP-kazrin C and GFP-kazrin C-Nt *foci* with TxR-Tfn loaded endosomes; and *Figure 4—figure supplement 4* and *Video 4* for co-localization of GFP-kazrin C with Eps15 homology domain (EHD) proteins. (**F**) Confocal micrographs of kazKO cells expressing low amounts (see M & M) of GFP, GFP-kazrin C, or GFP-kazrin C-Nt loaded with 20 µg/ml of TxR-Tfn at 16 °C and chased for 10 min at 37 °C. Cell borders are indicated by dashed lines and the nuclei in blue. (**G**) Scattered plots of the mean ± SD TxR-Tfn perinuclear enrichment for the cells and experimental conditions described in D. See M & M for experimental details. The data is normalized to the mean value of kazKO cells expressing GFP. *p*-values of the two-tailed Mann-Whitney test are shown. n>25 cells for each sample.

The online version of this article includes the following source data and figure supplement(s) for figure 4:

**Source data 1.** Un-cropped blots for *Figure 4A*.

**Source data 2.** Data for graphs presented in *Figure 4B, E and G*.

**Figure supplement 1.** Expression of GFP-kazrin C-Nt in kazKO MEF.

**Figure supplement 1—source data 1.** Un-cropped blots for *Figure 4—figure supplement 1*.

**Figure supplement 2.** Kazrin C co-localizes with p120 and β-catenins and N-cadherin at the plasma membrane (PM) and internal structures.

**Figure supplement 3.** Deletion of the C-terminal predicted intrinsically disordered region (IDR) reduces association of kazrin C with endosomes.

**Figure supplement 3—source data 1.** Data for graph presented in *Figure 4—figure supplement 3B*.

**Figure supplement 4.** GFP-kazrin C forms condensates associated with endosomal Eps15 Homology Domain (EHD)-protein-enriched subdomains.

---

that kazrin might also interact with the dynein/dynactin complex and serve as a candidate dynein/dynactin endosomal adaptor. Consistent with this hypothesis, we found that moderately expressed GFP-kazrin C pulled-down the dynactin component p150-glued from cell extracts as well as the dynein heavy chain, whereas GFP alone did not (*Figure 5H*). Similar to what has been described for other *bona fide* or candidate dynein/dynactin adaptors (*Celestino et al., 2022*; *Fenton et al., 2021*; *Kendrick et al., 2019*; *Schlager et al., 2010*; *Splinter et al., 2010*), we also detected co-immunoprecipitation of GFP-kazrin C with plus-end directed motors, specifically, with kinesin-1 (*Figure 5H*), a motor associated with EEs (*Loubéry et al., 2008*; *Schmidt et al., 2009*). We observed no co-immunoprecipitation with tubulin (*Figure 5H*), indicating that GFP-kazrin C interactions with dynactin, dynein, and kinesin-1 were not indirectly mediated by microtubules. Pull-down experiments with GST-kazrin C, expressed

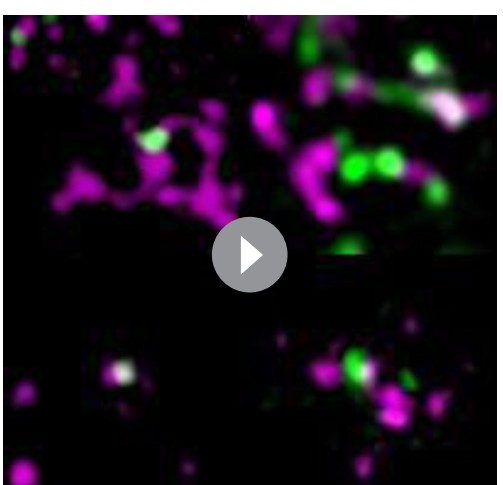

**Video 3.** Four 3D reconstructions of Z stacks of kazKO MEF expressing moderate amounts of GFP-kazrin C loaded with Texas Red-Tfn (TxR-Tfn) at 16°C to accumulate endocytic cargo in early endosome s (EEs). Cells were shifted to 37°C and fixed after 10 min. The windows are 5 × 5 µm$^2$.

https://elifesciences.org/articles/83793/figures#video3

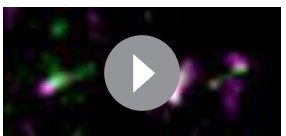

**Video 4.** Two 3D reconstructions of Z stacks of kazKO MEF expressing moderate amounts of GFP-kazrin C, fixed, and stained with α-EHD1 and A568-conjugated secondary antibodies. The windows are 5 × 5 µm$^2$.

https://elifesciences.org/articles/83793/figures#video4

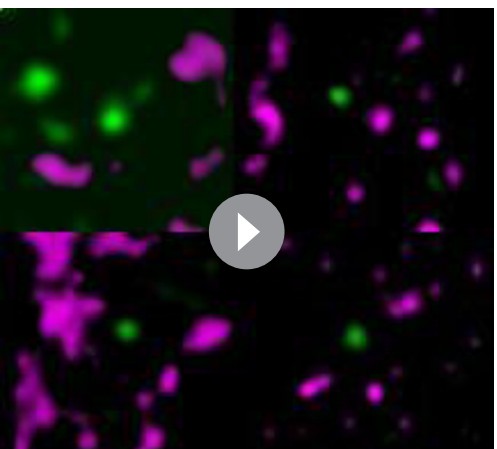

**Video 5.** Four 3D reconstructions of Z stacks of kazKO cells expressing moderate amounts of a GFP-kazrin C lacking the C-terminal predicted intrinsically disordered region (IDR) (GFP-kazrin C-Nt) loaded with TxR-Tfn at 16°C to accumulate endocytic cargo in early endosome (EE). Cells were shifted to 37°C and fixed after 10 min. The windows are 5 × 5 μm².
https://elifesciences.org/articles/83793/figures#video5

and purified from *E. coli,* and the dynactin complex, purified from pig (*Jha et al., 2017*), demonstrated that the interaction was direct and that it was contributed by both, the N- and C-terminal halves of kazrin C (*Figure 5I*), suggesting multiple contacts with different dynactin components. As also described for other dynein/dynactin adaptors (*Gonçalves et al., 2019*; *Horgan et al., 2010*; *Lee et al., 2020*; *Redwine et al., 2017*; *Schroeder and Vale, 2016*), pull-down experiments with purified components showed a specific interaction of kazrin C with one of the dynein LIC, in particular LIC1 (*Figure 5J*). Finally, immunoprecipitation experiments from MEFs using the polyclonal antibody against the C-terminus of kazrin C also suggested binding of endogenous kazrin with the dynactin complex (*Figure 5K*).

To test if GFP-kazrin C might be able to convey cargo other than endosomes to the pericentriolar region, we constructed a chimera bearing the mitochondrial targeting domain (MTD) of a testis-specific *D. melanogaster* centrosomin (*Chen et al., 2017*) (GFP-kazrin C-MTD), that lacks the centrosome targeting sequences but still bears the CM1 gamma-tubulin nucleating domain present in all centrosomin variants. This splice variant thereby converts mitochondria to microtubule organizing centers (MTOC) in spermatids (*Chen et al., 2017*). The GFP-kazrin C-MTD chimera bearing only the MTD but not the CM1 domain was transfected in Cos7 cells and the distribution of mitochondria was analyzed and compared to that of cells expressing GFP-kazrin C or GFP alone. As shown in *Figure 5—figure supplement 2*, GFP-kazrin C-MTD but not GFP or GFP-kazrin C, significantly increased the pericentriolar accumulation of mitochondria.

Our data indicated that kazrin might act as a candidate endosomal dynein/dynactin adaptor, with its C-terminal IDR working as a scaffold that entraps EEs or certain EE subdomains through multiple low-affinity binding sites. To test this hypothesis, we applied high-speed live-cell fluorescence imaging to visualize the movement of TxR-Tfn-loaded EEs in WT and kazKO cells. As previously described, EEs in WT cells were highly motile exhibiting long-range trajectories of several micrometers, followed by more confined movements within a 1 μm radius (*Driskell et al., 2007*; *Flores-Rodriguez et al., 2011*; *Loubéry et al., 2008*; *Nielsen et al., 1999*; *Rogers et al., 2010*; *Zajac et al., 2013*; *Video 8* (WT) and *Video 9*). Time projections of Z-stacks of 90 s Videos evidenced the linear endosomal trajectories in WT cells, with an average length of about 5 μm (*Figure 6A and B*; *Figure 6—figure supplement 1* and *Video 8* (WT) and *Video 9*). However, in kazKO MEFs, the kymographs showed a profusion of bright dots as compared to the straight trajectories in the WT, and the length of the straight trajectories (longer than 1 μm) was significantly reduced (*Figure 6A and B*; *Figure 6—figure supplement 1* and *Videos 8–10*). These observations suggested that the absence of kazrin reduced the association of EEs with some microtubule-dependent motors and/or diminished their processivity or velocity. Analysis of the maximum instantaneous velocities (Vi) of retrograde trajectories longer than 1 μm, mostly contributed by dynein (*Flores-Rodriguez et al., 2011*; *Loubéry et al., 2008*), showed that those were lower in the kazKO cells, as compared to the WT (*Figure 6C* and *Videos 8–10*). Finally, and also supporting the view that kazrin directly contributes to EE retrograde transport, we observed that expression of GFP-kazrin C, but not expression of the truncated version lacking the C-terminal endosomal binding region (GFP-kazrin C-Nt) nor GFP alone, rescued the endosome motility defects installed by depletion of kazrin (*Figure 6A* to *C*; *Figure 6—figure supplement 2* and *Videos 11–14*).

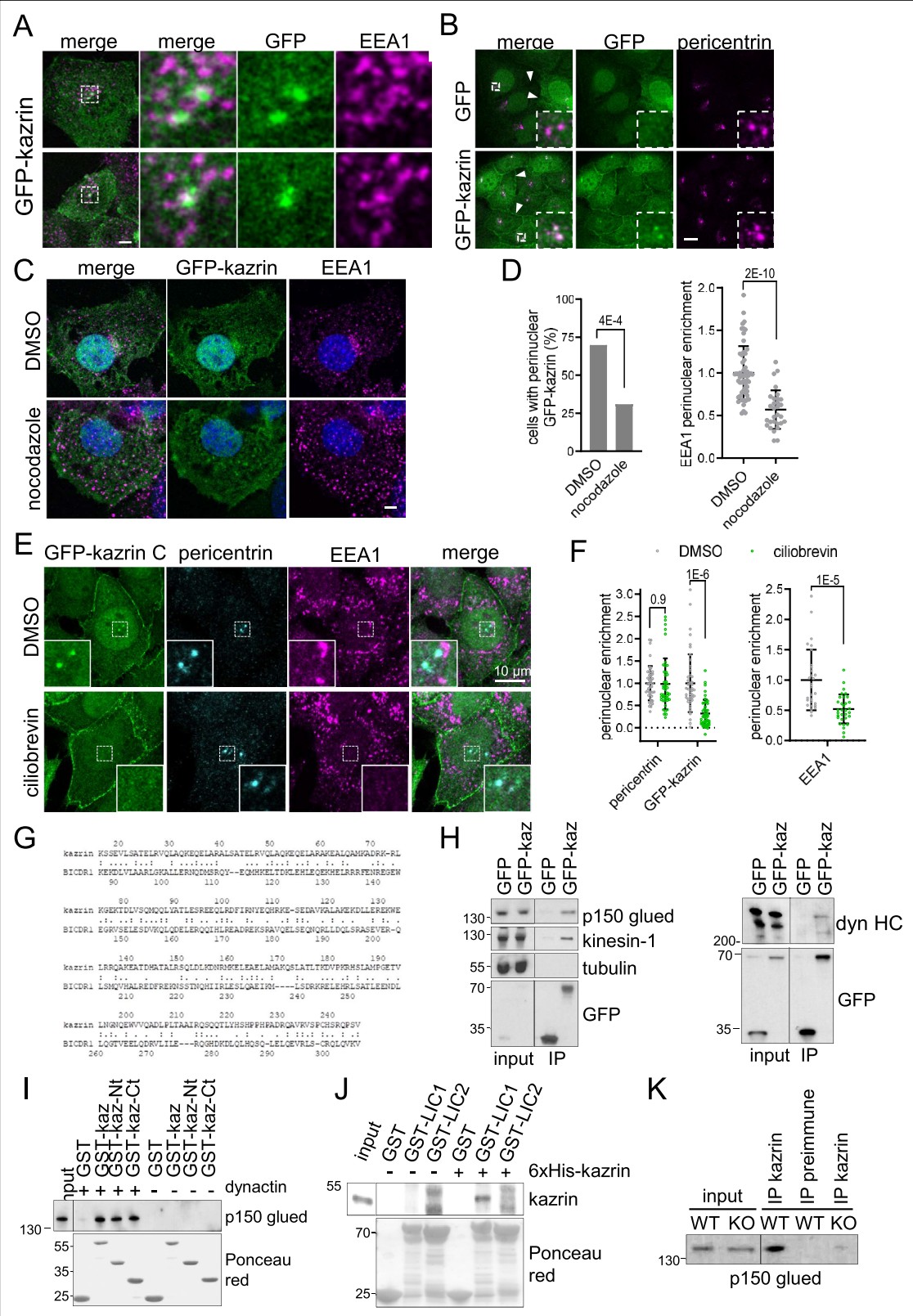

**Figure 5.** Karin C concentrates in the pericentriolar region and interacts with dynactin and dynein. (**A**) Merged confocal fluorescence micrographs of kazKO MEF moderately expressing (See M & M) GFP-kazrin C, fixed and stained with anti-EEA1 and A568-conjugated secondary antibodies. Individual channels and merged images of 6 x magnifications are shown. Scale bar, 10 μm. (**B**) Merged and individual channels of confocal fluorescence micrographs of kazKO MEF moderately expressing GFP or GFP-kazrin C, fixed and stained with anti-pericentrin and A568-conjugated secondary

*Figure 5 continued on next page*

*Figure 5 continued*

antibodies. 3.5 x magnifications are shown. Arrowheads indicate cell-cell borders. Scale bar, 10 μm. Refer also to *Figure 5—figure supplement 1* and *Videos 6 and 7* for live imaging. (**C**) Confocal fluorescence micrographs of kazKO MEF moderately expressing GFP-kazrin C, treated with DMSO or 100 ng/ml nocodazole, fixed, and stained with anti-EEA1 and A568-conjugated secondary antibodies and DAPI. Scale bar, 5 μm. (**D**) Bar plot showing the percentage of cells with a perinuclear localization of GFP-kazrin C (left) and scattered plot of the mean ± SD EEA1 perinuclear enrichment (right) in the cells and the experimental conditions described in C. See M & M for further details. The data was normalized to the mean of the mock-treated cells. *p*-values of a two-sided Fisher's exact test (left) and a two-tailed Student *t*-test (right) are shown. n>32 cells for each sample. (**E**) Confocal fluorescence micrographs of kazKO MEF moderately expressing GFP-kazrin C, treated with DMSO or 40 nM ciliobrevin, fixed, and stained with mouse anti-EEA1 and rabbit anti-pericentrin antibodies and A488 and A568-conjugated secondary antibodies, respectively, and DAPI. 5.4 × 5.4 μm² magnified areas where the pericentrin *foci* accumulate are shown. (**F**) Scattered plot of the mean ± SD intensity signal of pericentrin and GFP-kazrin C in the pericentriolar dots (left), for the cells and experimental conditions described in E, normalized to the mean signal of pericentrin or GFP-kazrin C, in cells treated with DMSO. Scattered plot of the mean ± SD EEA1 perinuclear enrichment (right) in cells treated as described in E. See M & M for further details. The data was normalized to the mean of the mocktreated cells. *p*-values of the two-tailed Student *t*-test are shown. n>44 cells for each sample. (**G**) Sequence comparison between kazrin C and human BICDR1 (LALIGN). (**H**) Immunoblots of anti-GFP agarose immunoprecipitates (IP) from cell lysates of kazKO MEF moderately expressing GFP or GFP-kazrin C, probed for the indicated proteins. (**I**) Immunoblots of pull-downs with of glutathione-Sepharose beads coated with purified GST or GST fused to kazrin C (GST-kazrin) or its N- (amino acids 1–176) (GST-kazrin-Nt) or C-terminal (amino acids 161–327) (GST-kazrin-Ct) portions, incubated with (+) or without (−) dynactin complex, purified from pig. The membranes were probed with an anti-p150 glued antibody or stained with Ponceau red to detect the GST constructs. (**J**) Immunoblots of pull-downs with glutathione-Sepharose beads coated with GST, GST-LIC1, or GST-LIC2, incubated with purified 6xHis-kazrin C. The membranes were probed with a mouse anti-kazrin antibody or stained with Ponceau red to detect the GST constructs. (**K**) Immunoblot of protein A-Sepharose immunoprecipitates (IP) from WT or kazKO MEFs using a mix of rabbit polyclonal serums against the N- and C-terminal domains of kazrin C or a pre-immunization serum, probed with an anti-p150 glued (dynactin) antibody. The low amounts of endogenous kazrin could not be detected in the immunoprecipitates with any of the antibodies tested because the antibody had a molecular weight similar to endogenous kazrin (about 50 Kda) and interfered with the detection. The kazKO MEF was used as a specificity control instead.

The online version of this article includes the following source data and figure supplement(s) for figure 5:

**Source data 1.** Un-cropped blots for *Figure 5H,I, J and K*.

**Source data 2.** Data for graphs presented in *Figure 5D and F*.

**Figure supplement 1.** GFP-Kazrin C forms *foci* that travel toward the pericentriolar region.

**Figure supplement 2.** GFP-Kazrin C fused to the mitochondrial targeting domain (MTD) of centrosomin conveys mitochondria to the pericentriolar region.

**Figure supplement 2—source data 1.** Data for graphs presented in *Figure 5—figure supplement 2B*.

# Discussion

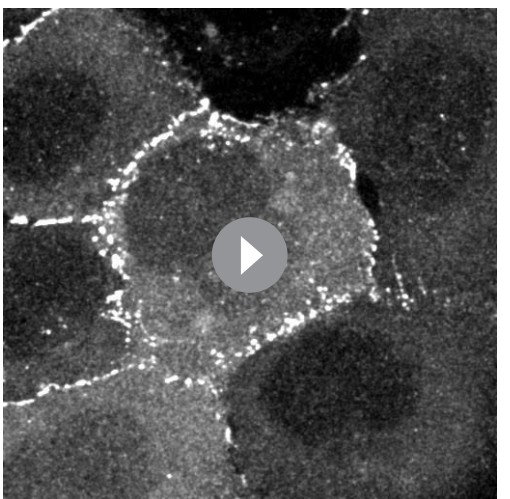

**Video 6.** 2.65 s time-lapse video of the juxtanuclear region of a kazKO MEF moderately expressing GFP-kazrin C (GFP-kaz) with a confocal microscopy. A 51.2 x 51.2 μm² window is shown.

https://elifesciences.org/articles/83793/figures#video6

Overall, our data suggest the kazrin plays a primary role in endosomal recycling through the long pathway traversing the pericentriolar, juxta-nuclear region, and that it does so by promoting dynein-dependent retrograde transport of EEs or EE-derived profiles in transit to the REs. We showed that the predicted IDR of kazrin directly interacted with several EE components implicated in endocytic traffic (*Caplan et al., 2002*; *George et al., 2007*; *Grant et al., 2001*; *Grant*

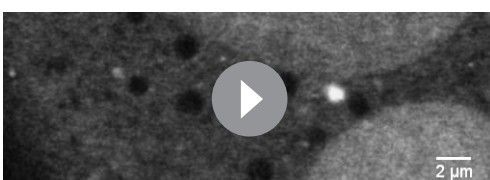

**Video 7.** 2.65 seconds time-lapse video of the juxtanuclear region of a kazKO MEF moderately expressing GFP-kazrin C (GFP-kaz) with a confocal microscopy. Scale bar, 2 μm.

https://elifesciences.org/articles/83793/figures#video7

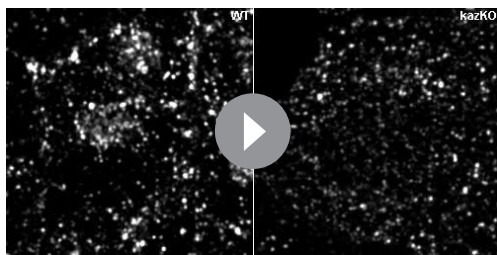

**Video 8.** 3 s time-lapse live-cell videos showing Texas Red-Tfn (TxR-Tfn) loaded endosomal dynamics in WT and kazKO MEF. The windows are 42.5 × 42.5 μm². Cells were loaded with TxR-Tfn at 16°C to accumulate endocytic cargo in early endosomes (EEs) and imaged immediately after the shift to 37°C. The images correspond to the maximum intensity Z projection.
https://elifesciences.org/articles/83793/figures#video8

*and Caplan, 2008*; *Jović et al., 2007*; *Lin et al., 2001*; *Naslavsky et al., 2006*; *Perrin et al., 2013*; *Rapaport et al., 2006*) and that depletion of kazrin caused a delay in the transport of endocytosed Tfn to the juxtanuclear region, as well as the scattering of EEs but not REs to the cell periphery. These phenotypes recapitulate those observed when depleting other proteins involved in EE to RE transport such as EHD3, or upon inhibition of dynein (*Burkhardt et al., 1997*; *Driskell et al., 2007*; *Naslavsky et al., 2006*; *Nielsen et al., 1999*). We also showed that kazrin shared considerable homology with dynein/dynactin adaptors and that depletion or deletion of its IDR impairs the movement of Tfn-loaded endosomes toward the juxtanuclear region.

While the vesicular nature of membrane traffic from the PM to the EEs has been well characterized, the principles governing the transport of membranes and cargo within the endosomal system are much less understood. The more accepted view is that the core of the EEs, receiving the internalized material, undergoes a maturation process that leads to its conversion to LEs (*Huotari and Helenius, 2011*; *Podinovskaia and Spang, 2018*; *Wang et al., 2019*), while retrograde traffic is driven by tubular transport intermediates, generated by sortinexins (SNX) or clathrin-coated vesicles (*Briant et al., 2020*; *McNally and Cullen, 2018*). In addition, retrograde transport of EEs to the pericentriolar region has been proposed to facilitate fusion with or maturation to REs (*Naslavsky and Caplan, 2018*; *Solinger et al., 2020*).

Within this wide range of endosomal trafficking events, microtubules seem to play key roles. EEs move along microtubule tracks with a bias toward the cell center (*Driskell et al., 2007*; *Flores-Rodriguez et al., 2011*; *Nielsen et al., 1999*). Retrograde movement is mainly effected by the dynein/dynactin complex (*Driskell et al., 2007*; *Nielsen et al., 1999*; *Zajac et al., 2013*). Treatment of cells with nocodazol, or interfering with dynein, results in the inhibition of endosome motility, the scattering of EEs to the cell periphery, and impaired endosomal maturation (*Driskell et al., 2007*; *Flores-Rodriguez et al., 2011*; *Lam et al., 2010*; *Palmer et al., 2009*; *Valetti et al., 1999*; *Zajac et al., 2013*). In addition, plus and minus-end directed microtubule-dependent-motors have both a role in maintaining the endosomal subdomain organization and in the formation and motility of SNX-dependent tubular structures (*Hunt et al., 2013*; *Soppina et al., 2009*; *Traer et al., 2007*; *Wassmer et al., 2009*). Interestingly, the motility of different SNX tubules or endosomal subdomains is associated with distinct dynein complexes bearing either the LIC1 or LIC2 chains and particular kinesin types (*Hunt et al., 2013*). In this context, the interactome of kazrin C suggests that it might work as a LIC1-dynein and kinesin-1 candidate adaptor for EHD and/or AP-1/clathrin transport intermediates emanating from EEs, in transit to the RE. Hook proteins, as components of FHF (Fused Toes-Hook-Fused toes and Hook Interacting Protein) complexes, also work as EEs dynein/dynactin adaptors in yeast, fruit flies, and mammalian cells (*Bielska et al., 2014*; *Christensen et al., 2021*; *Krämer and Phistry, 1996*; *Lu et al., 2020*; *Olenick et al., 2019*; *Villari et al., 2020*; *Yao et al., 2014*; *Zhang et al., 2014*). However, the interactome of the mammalian hook1 and hook3 and the endocytic pathways affected by interfering with their function differ from those of kazrin and suggest that hook proteins promote motility of EEs subdomains in transit to late endosomes (LEs) rather than to REs (*Christensen et al., 2021*; *Guo et al., 2016*; *Maldonado-Báez and Donaldson, 2013*; *Olenick et al., 2019*; *Xu et al., 2008*). Likewise,

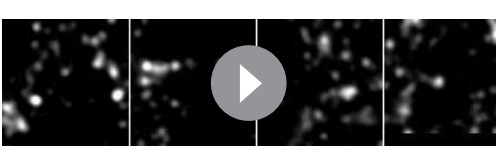

**Video 9.** Four 3s time-lapse live-cell videos showing Texas Red-Tfn (TxR-Tfn) loaded endosomal dynamics in WT MEF. The windows are 12.8 × 12.5 μm². Cells were loaded with TxR-Tfn at 16°C to accumulate endocytic cargo in early endosomes (EEs) and imaged immediately after the shift to 37°C. The images correspond to the maximum intensity Z projection.
https://elifesciences.org/articles/83793/figures#video9

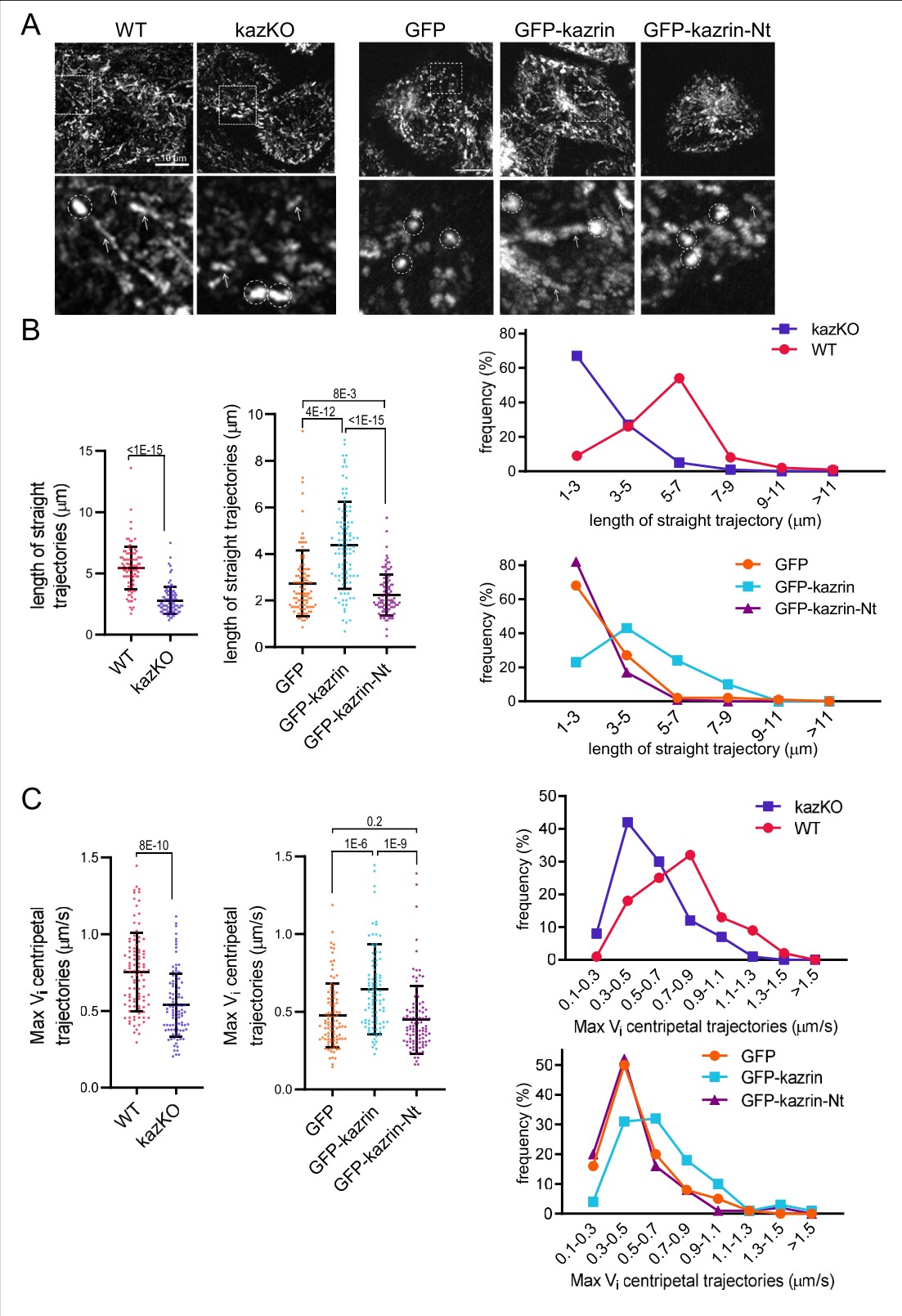

**Figure 6.** Depletion of kazrin impairs endosome motility. (**A**) Time projections of MIP of confocal fluorescence microscopy videos taken for 90 s with a 3 s time-lapse, of wild-type (WT) and kazKO MEF or kazKO MEF expressing low levels of GFP, GFP-kazrin C of a GFP-kazrin C construct lacking the C-terminal predicted intrinsically disordered region (IDR) (GFP-kazrin-Nt) (See M & M), showing trajectories of early endosome (EE) loaded with Texas Red-Tfn (TxR-Tfn) at 16°C. Cells were shifted to 37°C and immediately imaged. Scale bar = 10 μm. A magnified 10 × 10 μm² inset is shown below.

*Figure 6 continued on next page*

*Figure 6 continued*

Arrows point to straight trajectories and dashed circles indicate constrained endosome movements. (**B**) Scattered plots of the mean ± SD lengths of endosome trajectories (longer than 1 μm) (left graphs) for the cells and experimental conditions described in A. *p*-values of the two-tailed Mann-Whitney tests are shown. n=100 endosomes for each sample recorded in more than 20 cells. Line plots for the frequencies of the trajectory length in each cell type are shown on the right. (**C**). Scattered plots of the mean ± SD maximum instantaneous velocities (Vi) (left graphs) of retrograde endosome trajectories (longer than 1 μm) for the cells and experimental conditions described in A. *p*-values of the two-tailed Mann-Whitney tests are shown. n=100 endosomes for each sample recorded in more than 20 cells. Line plots for the frequencies of the maximum Vi for each cell type are shown on the right. See also *Video 8* for life imaging of an example of WT and kazKO cells loaded with TxR-Tfn, and *Videos 9 and 10* for four different magnified fields showing endosome motility in different WT and kazKO cells, respectively. See also *Video 11* for an example of life imaging of kazKO cells expressing low levels of either GFP, GFP-kazrin C, or GFP-kazrin C-Nt loaded with TxR-Tfn, and *Videos 12–14* for four different magnified fields showing endosome motility in these cell types.

The online version of this article includes the following source data and figure supplement(s) for figure 6:

**Source data 1.** Data for graphs are presented in *Figure 6B and C*.

**Figure supplement 1.** Depletion of kazrin impairs endosome motility.

**Figure supplement 2.** Depletion of kazrin impairs endosome motility.

the phenotypes installed by depletion of FIP3-RAB11, another dynein/dynactin adaptor connecting early and recycling endosomes (*Horgan et al., 2010*), do not really mimic those observed upon kazrin C deletion. The observations that FIP3-RAB11 mainly co-localizes with RE rather than EEs markers and that its depletion disperses the RAB11 compartments rather than the EEs (*Hales et al., 2001*; *Horgan et al., 2007*; *Horgan et al., 2004*), as opposed to kazrin depletion, suggest that FIP3-RAB11 plays a role downstream of kazrin. FIB3-RAB11 might either recieve EE/RE transport intermediates that have already acquired RAB11, or it could position the RE compartment at the pericentriolar region, which might be key for the docking and fusion of incoming transport intermediates.

How kazrin C promotes EEs retrograde motility at the molecular level is still an open question. Its homology to *bona fide* activating adaptors, its domain organization, and its interactome suggests that it might actually work by stabilizing the dynein/dynactin complex and by linking it to EEs, either alone or in concert with other cellular components. However, kazrin might as well work by promoting the kinesin-1-dependent transport of dynactin and/or dynein to the cell surface, which in turn is required for retrograde EE motility (*Abenza et al., 2009*; *Lenz et al., 2006*; *Zekert and Fischer, 2009*).

The role of kazrin in endocytic recycling might explain some of the pleiotropic effects observed in vertebrate development upon its depletion. Altered cell adhesion in *Xenopus* embryos and human keratynocytes (*Cho et al., 2010*; *Sevilla et al., 2008a*; *Sevilla et al., 2008b*) might derive from defective recycling of cadherins or desmosomal components (*Cadwell et al., 2016*; *Calkins et al., 2006*; *Demlehner et al., 1995*; *Kawauchi, 2012*; *Le et al., 1999*; *Lock and Stow, 2005*; *Roeth et al., 2009*; *Yan et al., 2016*). Indeed, depletion of kazrin in *Xenopus laevis* leads to decreased levels of E-cadherin, which can be reverted by inhibiting endocytic uptake (*Cho et al., 2010*). This observation is consistent with the role of kazrin diverting traffic of internalized E-cadherin away from the lysosomal compartment and back to the PM. Likewise, eye and craniofacial defects associated with reduced EMT and neural crest cell migration (*Cho et al.,*

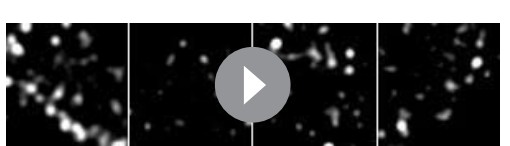

**Video 10.** Four 3 s time-lapse live-cell videos showing Texas Red-Tfn (TxR-Tfn) loaded endosomal dynamics in kazKO MEF. The windows are 12.8 × 12.5 μm². Cells were loaded with TxR-Tfn at 16°C to accumulate endocytic cargo in early endosomes (EEs) and imaged immediately after the shift to 37°C. The images correspond to the maximum intensity Z projection.
https://elifesciences.org/articles/83793/figures#video10

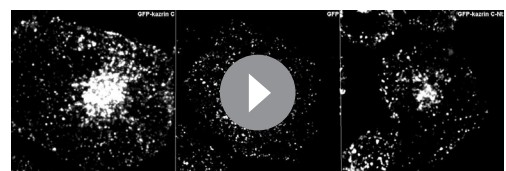

**Video 11.** 3 s time-lapse live-cell videos showing Texas Red-Tfn (TxR-Tfn) loaded endosomal dynamics in kazKO MEF expressing low levels (See M&M) of GFP, GFP-kazrin C, or GFP-kazrin Nt. The windows are 42.5 × 42.5 μm². Cells were loaded with TxR-Tfn at 16°C to accumulate endocytic cargo in early endosomes (EEs) and imaged immediately after the shift to 37°C. The images correspond to the maximum intensity Z projection.
https://elifesciences.org/articles/83793/figures#video11

**Video 12.** Four 3 s time-lapse live-cell videos showing Texas Red-Tfn (TxR-Tfn) loaded endosomal dynamics in kazKO MEF expressing low levels (See M & M) of GFP-kazrin C. The windows are 12.8 × 12.5 μm². Cells were loaded with TxR-Tfn at 16°C to accumulate endocytic cargo in early endosomes (EEs) and imaged immediately after the shift to 37°C. The images correspond to the maximum intensity Z projection.
https://elifesciences.org/articles/83793/figures#video12

**Video 14.** Four 3 s time-lapse live-cell videos showing Texas Red-Tfn (TxR-Tfn) loaded endosomal dynamics in kazKO MEF expressing low levels (See M & M) of GFP-kazrin C-Nt. The windows are 12.8 × 12.5 μm². Cells were loaded with TxR-Tfn at 16°C to accumulate endocytic cargo in early endosomes (EEs) and imaged immediately after the shift to 37°C. The images correspond to the maximum intensity Z projection.
https://elifesciences.org/articles/83793/figures#video14

*2011*), might originate from altered endocytic trafficking of integrins, cadherins, and/or signaling receptors (*Cadwell et al., 2016*; *Jones et al., 2006*; *Wilson et al., 2018*).

It is worth noticing that kazrin is only expressed in vertebrates, whose evolution is linked to an explosion in the number of cadherin genes and the appearance of desmosomes (*Green et al., 2020*; *Gul et al., 2017*). In this context, it is tempting to speculate that while the core machinery involved in membrane traffic is largely conserved from yeast to humans, vertebrates might have had the need to develop specialized trafficking machinery to spatiotemporally regulate the function of particular adhesion complexes. Therefore, kazrin might turn out to be a valid therapeutic target to selectively modulate the function of those adhesion complexes in the context of a myriad of human pathologies (*Kaszak et al., 2020*; *Yuan and Arikkath, 2017*). Identification of the relevant endocytic cargo traveling in a kazrin-dependent manner will be the next step to further understand the molecular, cellular, and developmental functions of kazrin.

# Materials and methods
## DNA techniques and plasmid construction
Oligonucleotides used for plasmids construction and information about the construction strategies are available upon request. DNA manipulations were performed as described (*Sambrook et al., 1989*), or with the Getaway cloning system (Life Technologies) in the case of lentiviral vectors. Enzymes for molecular biology were obtained from New England Biolabs. Plasmids were purified with the Nucleospin plasmid purification kit (Macherey-Nagel 740422.10). Linear DNA was purified from agarose gels using the gel extraction kit from Qiagen. Polymerase chain reactions (PCRs) were performed with the Expand High Fidelity polymerase (Roche) and a TRIO-thermoblock (Biometra GmbH). Plasmids used are listed in *Table 1*. *E. coli* DH5α (*Chan et al., 2013*) was used to amplify plasmids. All plasmids generated in this work are available for non-commercial purposes under request.

## Cell culture and cell line establishment

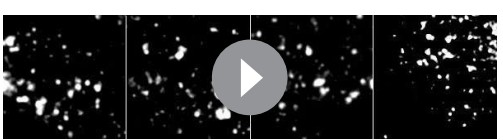

**Video 13.** Four 3 time-lapse live-cell videos showing Texas Red-Tfn (TxR-Tfn) loaded endosomal dynamics in kazKO MEF expressing low levels (See M & M) of GFP. The windows are 12.8 × 12.5 μm². Cells were loaded with TxR-Tfn at 16°C to accumulate endocytic cargo in early endosomes (EEs) and imaged immediately after the shift to 37°C. The images correspond to the maximum intensity Z projection.
https://elifesciences.org/articles/83793/figures#video13

Cos7 cells were obtained from the German Collection of Microorganisms and Cell Cultures (https://www.dsmz.de/dsmz) (ACC-60). It is a cell line with fibroblast features derived from CV-1, a simian (Cercopithecus aethiops) cell line, by transformation with an origin-defective mutant of SV-40; cells were described in the literature to support the growth of SV-40 viruses; classified as risk category 1 according to the German Central Commission for Biological Safety (ZKBS). Mice embryonic fibroblast (MEF) was provided by S. Offermanns, (University of Heidelberg, Germany) and derived from C57BL/6 mice (*Offermanns et al., 1997*). mIMCD3 were kindly provided

**Table 1.** Plasmids.

| Plasmid | Insert | Backbone |
|---|---|---|
| pGEX-5X-3 | GST | pGEX-5X-3 |
| pGST-hB24 | GST + kazrin C, human gene KIAA1026 | pGEX-4T-2 |
| pGST-kaz-Ct (161-327) | GST + kazrin C Ct (aa 161–327) | pGEX-5X-3 |
| pGST-kaz-Nt (1-176) | GST + kazrin C Nt (aa 1–176) | pGEX-5X-3 |
| pGST-kaz-Ct-KA | GST + kazrin C Ct (aa 161–327) -(281-KRKKKK-286, AAAAAA) | pGEX-5X-3 |
| pQE11-kazrin | 6xHis + kazrin C | pQE11 |
| pGST-EHD1 | GST + EHD1 | pGEX-5X-3 |
| pGST-EHD3 | GST + EHD3 | pGEX-5X-3 |
| pGST-γ-Adaptin-ear | GST + human AP1 Adaptin G1 ear (aa 702–925) | pGEX-5X-3 |
| pGST-CHC17-TD | GST + human CHC17-aa1-483 (CHC TD +linker) | pGEX-5X-3 |
| pGST-LIC1 | GST + dynein light intermediate chain 1 | pGEX-5X-3 |
| pGST-LIC2 | GST + dynein light intermediate chain 2 | pGEX-5X-3 |
| pX458-kaz KO 1 | Cas9 and Cas9 target sequence 1 | pSpCas9(BB)–2A-GFP (pX458) |
| pX458-kaz KO 2 | Cas9 and Cas9 target sequence 2 | pSpCas9(BB)–2A-GFP (pX458) |
| pVSV-G | Lentivirus envelope protein | pLenti-CMV |
| pAX8 | Lentivirus packaging protein | pLenti-CMV |
| pINDUCER-EGFP | EGFP | pINDUCER20 |
| pINDUCER-EGFP-kazrin C | EGFP + kazrin C | pINDUCER20 |
| pINDUCER-EGFP-kazrin C-Nt | EGFP + kazrin C (aminoacids 1–176) | |
| pKLO.1_sh*Kzrn* | cloneID TRCN00001 82832 | pLK0.1 |
| pLK0.1 | SHC002 | pLK0.1 |
| pCMV-dR8.2dvpr | | |
| pCMV-VSG-G | | |
| pEGFP-C2 | | |
| pEGFP-kazrin C | pEGFP-C2 + kazrin C | pEGFP-C2 |
| pEGFP-kazrin C-MTD | pEGFP-C2 + kazrin C fused to the Mitochondrial Targeting domain of *D. melanogaster* centrosomin CnnT splice variant (AT9084) (aa 212–480) | pEGFP-C2 |

by F. García-Belmonte from the Centro de Biología Molecular Severo Ochoa (Spain) and they were purchased from the American Type Culture Collection ATCC (CRL-2123). mIMCD-3 is an inner medullary collecting duct (IMCD) cell line derived by Michael Rauchman from a mouse transgenic for the early region of SV40 [Tg(SV40E)bri/7]. It has epithelial morphology. MEF and Cos7 cells were grown in DMEM (Thermo Fisher Scientific, 21969035) supplemented with 10% FBS, 100 μ/ml penicillin, 100 μg/ml streptomycin, and 2 mM L-glutamine (Thermo Fisher Scientific, 25030081) in a humidified 5% $CO_2$ atmosphere at 37 °C. mIMCD3 were grown in DMEM/F-12 (Thermo Fisher Scientific, 21331020) with 10% FBS. Cos7 were transiently transfected with Lipofectamine 2000 (Thermo Fisher Scientific,

11668027). Cells were analyzed 24 hr after transfection. For shRNA kazrin depletion, pKLO.1_sh*Kzrn* from Merck Mission Library 2007 (Clone ID TRCN000018283) was used. pLKO.1_CV/_SCR (SHC002) was used as a control. For lentivirus production and Cos7 cell transfection, HEK293T cells were co-transfected with either the pLL3.7 encoding GFP, for virus production control and infection efficiency monitoring, or with pLKO.1 encoding the desired shRNA, and the viral packaging (pCMV-dR8.2 dvpr) and envelope (pCMV-VSV-G) plasmids, using calcium phosphate transfection. About 16 hr after transfection, the medium was changed and half of the usual volume was added. During the two following days, the medium containing the virus was collected and filtered with a 0.45 µm filter. The filtered virus solution was directly used for the infection of cell lines or stored in aliquots at –80 °C without prior concentration of the virus. Infection and selection of stably infected cells were done in the presence of the appropriate concentration of puromycin, titrated by using the minimum antibiotic concentration sufficient to kill untransfected cells, but to maintain cells transfected with the pLL3.7 GFP-containing plasmid. Actual depletion of kazrin or the protein of interest was analyzed by immunoblot using home-made polyclonal rabbit antibodies raised against the N- (amino acids 1–176) and C- terminal (amino acids 161–327) portions of kazrin C.

MEF and mIMCD3 KO cells were produced with the CRISPR-Cas9 system. Two guide RNAs were designed to recognize regions at the start of exon 2 of the *Kzrn* gene, corresponding to the start of kazrin C isoform (CACCGAATGCTGGCGAAGGACCTGG and CACCGCCTTCTGTACCAGCTGCACC). Online tools Benchling (https://www.benchling.com/) and the Broad Institute tool GPP (available here) were used for the design. Guide RNA oligonucleotides were annealed and inserted into a pSpCas9(B-B)–2A-GFP pX458 vector. MEFs were electroporated with Nucleofector (Lonza), following the manufacturer's instructions. GFP-positive cells were sorted by FACS in an Aria FUSION (Becton Dickinson) sorter and screened by immunoblotting with antibodies against the N-terminal and the C-terminal portions of kazrin. Lentiviral particles were produced in HEK 293T cells. Calcium phosphate-mediated transfection was used to deliver vector pINDUCER20 encoding GFP or GFP-tagged kazrin constructs, together with packaging and envelope lentiviral vectors. The supernatant of transfected HEK 293T cells was collected after 16 hr, 0.45 µm-filtered, and added to MEFs. The cells were passaged for a week, and incubated with 5 µg/ml doxycycline for 48 hr to induce the expression of the construct. GFP-positive cells were selected by FACS and pooled. MEFs were transfected by electroporation using the Ingenio solution (Mirus, MIR50108) and a nucleofector (Amaxa). The cells were processed 2 days after electroporation.

For complementation assays, GFP, GFP-kazrin C, and GFP-kazrin C-Nt were induced for up to 12 hr to achieve low, nearly-physiological expression levels of GFP-kazrin C (as compared to endogenous kazrin by immunoblot, using the home-made rabbit polyclonal anti-kazrin serums), and analogous expression levels of GFP or GFP-kazrin C-Nt (as compared by immunoblot using the mouse anti-GFP antibody (see antibodies section)). For GFP-kazrin C imaging or biochemical studies, cells were induced for up to 24 hr to achieve analogous, moderately-overexpressed levels of the proteins. To study the effect on microtubule dynamics and dynein inhibition, MEFs were treated with 100 ng/ml of nocodazole for 16 h or 40 nM of ciliobrevin for 12 hr, respectively, or DMSO, and then fixed at room temperature. kazKO MEF and kazKO MEF expressing GFP and GFP-kazrin C are available for non-commercial purposes under request. Cos7 and mIMCD3 we authenticated by ATCC. MEF was authenticated by proteomic analysis. All cell lines were tested negative for mycoplasma by PCR using EZ-PCR Mycoplasma detecting kit (VITRO, SA) or custom-made oligonucleotides.

## TxR-Tfn accumulation, juxtanuclear enrichment, and recycling assays

Cos7 cells or MEFs were grown on R-collagen-coated glass coverslips. For all assays, cells were starved 30 min in DMEM without FBS or bovine serum albumin (BSA). For the accumulation assays, cells were then incubated with pre-warmed DMEM containing 20 µg/ml of TxR-Tfn (from human serum, Molecular Probes, T2875) and 0.1% BSA for the specified times. Cells were washed in ice-cold PBS once and fixed in 4% PFA for 20 min on ice. For the TxR-Tfn juxtanuclear enrichment and recycling assays, 20 µg/ml of TxR-Tfn in DMEM with 0.1% BSA was added and cells were incubated at 16 °C for 30 min to load EEs. Cells were then washed in ice-cold PBS with 25 mM acetic acid pH 4.2, and with PBS and subsequently incubated with 500 µg/ml unlabeled Tfn (Merck, 616395) in DMEM with 0.1% BSA at 37 °C. Cells were then transferred to the ice at the indicated time points, washed in ice-cold PBS with 25 mM acetic acid pH 4.2 and with PBS, and fixed in 4% PFA for 20 min on ice. For the juxtanuclear

enrichment assays the mean TxR-Tfn fluorescence intensity within a 10 μm diameter circle in the juxta-nuclear region was divided by the signal in the whole cell selected with the Fiji free hand tool to define the ROI (Region of interest), at 10 min chase, after background subtraction. For recycling experiments, the mean fluorescence intensity per cell was measured using the Fiji free hand drawing tool to select the ROI at the indicated time points and the signal was normalized to the average intensity at time 0.

For TxR-Tfn accumulation, juxtanuclear enrichment, and recycling assays, images were taken with a Zeiss LSM780 confocal microscope equipped with a 63 x oil (NA = 1.4) objective, a GaAsP PMT detector 45% QE and images were acquired at pixel size 0.06 μm, unless otherwise indicated. For the experiments shown in 4 F, an Andor Dragonfly spinning disk microscope equipped with a 100 x oil (NA = 1.49) objective and a Sona 4.2 B11 sCMOS camera 95% QE was used. Images were acquired at pixel size 0.05 μm. At least two independent membrane traffic assays were performed with at least two biological replicas per experiment, with analogous results. Data from biological replicas from a representative experiment were combined to generate the graphs.

## GFP-kazrin and TxR-Tfn association analysis and immunofluorescence

3D reconstructions of EEs loaded with TxR-Tfn in cells expressing GFP-kazrin C or GFP-kazrin C-Nt were performed with voxel size 0.05 × 0.05 × 0.10 μm, compiled with the Andor Dragonfly spinning disk microscope equipped with a 100 x oil (NA = 1.49) objective and a Sona 4.2 B11 sCMOS camera 95% QE, in cells treated as for the TxR-Tfn recycling assay, immediately upon the shift from 16°C to 37°C. 3D videos of 5 × 5 μm² were generated with the Fiji 3D reconstruction tool. A 2.0 Gaussian blur filter was applied to the images after performing the 3D reconstruction and the contrast and bright-ness were modified to eliminate the cytosolic or nuclear background. Once the 3D reconstruction was built, the TxR *foci* (EE) closest to the GFP-kazrin C or GFP-kazrin C-Nt *foci* were identified manually by measuring with the line Fiji toll bar, and the distance between the centroid of those *foci* was measured with the same tool in the video frame showing the maximal separation. For immunofluorescence experiments**,** cells were seeded onto cover-glasses and fixed with 4% PFA in PBS containing 0.02% BSA and 0.02% sodium azide (PBS*), for 10 min at room temperature. Cells were washed three times for 5 min with PBS* and permeabilized with PBS* containing 0.25% Triton X-100 for 10 min. Cells were washed three times for 5 min with PBS* and incubated for 20 min in PBS* containing 1% BSA. Cells were then incubated in the presence of the primary antibody in PBS* for 1 hr at room temperature, washed three times with PBS*, and incubated for 1 hr in the presence of the secondary antibodies prepared in PBS*. Cells were washed three times with PBS* and mounted using Prolong Gold that included DAPI for nuclear staining (Thermo Fisher Scientific, P36934). Images were taken with a Zeiss LSM780 confocal microscope equipped with a 63 x oil (NA = 1.4) objective, a GaAsP PMT detector 45% QE, and images were acquired at pixel size 0.06 μm for the experiments shown in *Figure 5B and C* and 0.120 μm for the experiments shown in *Figure 5A* and 0.110 μm for the experiments in *Figure 5E*. Images shown in *Figure 4—figure supplement 4* and the associated videos for the 3D reconstruction of EHD labeled endosomes, were performed with the Andor Dragonfly spinning disk microscope equipped with a 100 x oil (NA = 1.49) objective and a Sona 4.2 B11 sCMOS camera 95% QE, with voxel size 0.05 × 0.05 × 0.10 μm. Experiments shown in *Figure 4—figure supplement 2* and *Figure 1—figure supplement 1* were acquired with a Leica TCS-SP5 confocal microscope equipped with a 63 x oil objective (NA = 1.4), with a pixel size of 0.06 μm. Juxtanuclear enrichments for EEA1 and RAB11 in MEFs were calculated after background subtraction as the mean fluorescence intensity within a 10 and 9 μm (respectively) diameter circle in the juxtanuclear region, divided by the mean intensity in the whole cell, as delimited with the Fiji free hand drawing tool to select the ROI. Association of GFP-kazrin C with pericentrin *foci* in the experiment shown in *Figure 5E and F* was measured by drawing a circle of 1.1 μm in diameter around the pericentrin *foci* and measuring the mean intensity for the pericentrin and GFP-kazrin C upon background subtraction. Data was normalized to the mean intensity of mock-treated cells.

## Cell migration and division assays

Cells were plated on 400 μg/ml Matrigel (Corning, CLS354234)-coated plates at low density and incubated for 5 hr. Once the cells were attached, the medium was replaced by Matrigel for 30 min to embed the cells in a matrix. Matrigel excess was then removed and cells were kept at 37 °C with 5% $CO_2$ during imaging. Phase contrast images were taken every 10 min for a total of 9 hr with a motorized

bright field Leica AF7000 microscope equipped with a 10 x objective (NA = 0.3), and a digital Hamamatsu ORCA-R2 CCD camera and images were taken with a pixel size of 0.64 µm. To analyze cell migration, cells were tracked using the Fiji plugin MTrackJ. Speed and direction persistency was calculated using the open-source program DiPer (*Dang et al., 2013*). To detect cytokinesis delay compatible with a defect in abscission, the time was measured from the moment daughter cells attach to the substrate until they completely detach from each other. At least two independent experiments with at least two biological replicas were performed for the motility and cell separation assays with analogous results. The data of different biological replicas of a representative experiment were combined to generate the graphs.

## Live confocal imaging and endosome motility analysis

Cells were seeded on plates with polymer coverslips for high-end microscopy (Ibidi, 81156). Cells were kept at 37 °C with 5% $CO_2$ during the imaging. For the *Videos 6 and 7* and the *Figure 5—figure supplement 1*, images were taken every 2.65 s on a Zeiss LSM780 confocal microscope equipped with a 63 x oil objective (NA = 1.4) with voxel size 0.05 × 0.05 × 0.130 µm. To follow EE motility, cells were starved for 30 min in DMEM without FBS and subsequently loaded at 16 °C with 20 µg/ml TxR-Tfn in DMEM with 0.1% BSA, as described for the TxR-Tfn recycling experiments. Cells were then rinsed with PBS and imaged immediately upon the addition of 37°C pre-warmed media loaded with unlabeled Tfn. Images were compiled with voxel size 0.17 × 0.17 × 0.46 µm for WT and KO cells and 0.09 × 0.9 × 0.46 µm for GFP GFP-kazrin C and GFP-kazrin C-Nt expressing cells, and they were taken every 3 s for 1.5 min using the Andor Dragonfly 505 microscope, equipped with a 60 x oil (NA = 1.4) objective and a Sona 4.2 B11 sCMOS camera 95% QE. Maximum intensity projections of the Z-stacks were generated with Fiji, after background subtraction and registration using the Linear Stack Alignment with SIFT tool of Fiji. Videos were generated from the original videos using the Fiji crop tool and a 1.0 Gaussian filter was applied. Kymographs of the maximum intensity Z-stack projections were generated to measure the length of linear trajectories with the Fiji free-hand line tool. Maximum instantaneous velocity (Vi) of TxR-Tfn loaded endosomes was measured by manually tracking endosomes moving into the cell center with the Fiji plugin MTrackJ. Two independent experiments with three biological replicas were performed to analyze the endosome motility giving analogous results. The data from different replicas of one of the experiments were combined to generate the graphs.

## SDS-PAGE and immunoblots

SDS–PAGE was performed as described (*Laemmli, 1970*), using pre-casted Mini-PROTEAN TGX 4–20% Acrylamide gels (Bio Rad, 4561096). Protein transfer, blotting, and chemiluminescence detection were performed using standard procedures. Detection of proteins was performed using the ECL kit (Cytiva, RPM2209).

## Cell fractionation

Cell fractionation was performed as described in *Li and Donowitz, 2014*. Briefly, cells were scraped from the plate, harvested by centrifugation at 700 g for 10 min, and resuspended in 1 ml of ice-cold Lysis Buffer (LB: 25 mM Hepes pH 7.4, 150 mM NaCl, 1 mM DTT, 2 mM EGTA) containing protease inhibitors. The cell suspension was then passed 10 times through a 27 G needle. The lysate was cleared by centrifuging twice at 3000 g for 15 min. The supernatant was subsequently centrifuged at 186,000 g for 1 hr at 4 °C to fractionate cellular membranes from the cytosol. The membrane pellet was resuspended in LB with protease inhibitors, passed 10 times though a 27 G needle, and laid on an Optiprep (Merck, 92339-11-2) gradient. A 12 ml 2% step Optiprep gradient in LB ranging from 32 to 10% was prepared beforehand in Ultra-Clear tubes (Beckman Coulter). Samples were spun for 16 hr at 100,000 g at 4 °C. 0.6 ml fractions were carefully collected from the top. Samples were then precipitated with trichloroacetic acid, air-dried, and resuspended in SDS-PAGE sample buffer for immunoblot analysis. For the experiments shown in *Figure 4A*, the supernatant from the 3000 g centrifugation was adjusted to 1 mg/ml of total protein and centrifuged at 186,000 g for 1 hr at 4 °C to fractionate cellular membranes (pellet) from the cytosol (supernatant). 15 µg of total protein from the 3000 g supernatant (total) and the corresponding one and five equivalents of the cytosolic or membrane fractions were loaded in an SDS-PAGE acrylamide gel and immunoblotted for EHD proteins or GFP. Cell fractionation assays in MEF were reproduced in at least three independent experiments. Confirmatory cell

fractionation experiments in Cos7 and IMDC cells were performed once. To quantify the ECL signal in *Figure 4A* the band intensities of the cytosolic and microsomal fractions were measured with Fiji using the rectangular drawing box, after background subtraction, in films with similar exposures for all samples. The microsomal fraction was divided by five and normalized by the total signal calculated as the cytosolic +5 x microsomal signals.

## GST pull-downs, GFP-trap, and endogenous immunoprecipitations

Purification of recombinant GST and 6-His fusion proteins from BL21 *E. coli* (Novagen, D48406) was performed as described (*Geli et al., 2000*). Pull down experiments were performed with Glutathione-Sepharose beads (GE Healthcare, 17-0756-01) coated with 0.5 µg of the indicated GST-tagged proteins and 2 nM of eluted 6xHis-kazrin C incubated in 1 ml of binding buffer containing PBS or 2 nM of the dynactin complex in 0.5 ml of DBB (25 mM Tris-HCl pH 8, 50 mM KoAc, 0.5 mM ATP, 1 mM DTT, 1 mM MgCl$_2$, 1 mM EGTA and 10% glycerol), both bearing 0.2% Triton-X100 and 0.5% BSA with protease inhibitors (Complete Roche, 11836145001), for 1 hr at 4 °C in a head-over-shoulder rotation. Beads were washed three times with the corresponding binding buffer containing Triton-X100 and twice without detergent. The beads were boiled in Laemmli buffer. Input and pulled-down samples were loaded in an SDS-PAGE gel and analyzed by immunoblot. For the pull downs from mammalian protein extracts, GST, and the GST-kazrin C N- (amino acids 1–176) and C-terminal (amino acids 161–327) portions were expressed and purified from *E. coli* as described above, using glutathione-Sepharose beads, and the beads were incubated with the 3000 g supernatant of a non-denaturing protein extract from WT MEF, prepared as described for the subcellular fractionation using LB, after adding 1% Triton-X100. After 1 hr incubation, beads were recovered and washed with LB 1% Triton-X100 three times and twice with LB buffer. Beads were resuspended in SDS-PAGE sample buffer and analyzed by immunoblot against EHD proteins and γ-adaptin.

For immunoprecipitations from MEFs, moderately overexpressing GFP and GFP-kazrin C, the cells were harvested and cleared at 100 g. The pellet was resuspended in 500 µl of IP buffer (20 mM Hepes, 50 mM KAc, 2 mM EDTA, 200 mM sorbitol, 0.1% Triton X-100, pH 7.2) containing protease inhibitors and passed 30 times through a 27 G needle. The lysate was cleared by centrifuging for 5 min at 10.000 g. 10 µl of GFP-binding agarose beads (Chromotek, gta-20) were incubated with the protein extract for 1 hr at 4 °C in head-over-shoulder rotation. Beads were washed six times with 1 ml of IP buffer. The beads were boiled in Laemmli buffer. Input and IP samples were loaded in an SDS-PAGE gel and analyzed by immunoblot. DBB (25 mM Tris-HCl pH 8, 50 mM KoAc, 0.5 mM ATP, 1 mM DTT, 1 mM MgCl$_2$, 1 mM EGTA, and 10% glycerol) containing 0.1% Triton X-100 was used for the immuno-precipitation experiments with dynactin and kinesin-1.

For endogenous immunoprecipitations, WT or kazKO cell extracts were generated as described above but incubated with rabbit IgGs against the kazrin C C-terminus (aa 161–327), pre-bound from a serum to Protein A-Sepharose, or IgGs from the pre-immune serum. The amount of endogenous kazrin in the immunoprecipitates could not be assessed because the IgGs interfered with the detection. Immunoprecipitation and pull-down assays were performed at least twice with the proper controls to discard possible background signals. Analogous results were obtained in all replicas. All key interactions were demonstrated with at least two different techniques.

## Lipid strip and liposome pelleting assays

Lipid strips (Echelon, P-6100) were incubated in 1% skimmed milk in PBS for 1 hr at room temperature. The corresponding GST fusion protein was added to a final concentration of 15 µg/ml in incubation buffer (10 mM Tris pH 8.0, 150 mM NaCl, 0.1% Tween-20, 3% BSA (fatty acid free, Merck, A7030)), with protease inhibitors over night at 4 °C. The strips were washed three times for 10 min in the incubation buffer and developed by immunoblot. Lipid strips with purified kazrin constructs were performed three times.

For the liposome pelleting assay, 1.2 µg of GST or GST-kazrin C expressed and purified from *E. coli* were dissolved in 100 µl of LBB (20 mM Hepes pH7.4, 120 mM NaCl, 1 mM EGTA, 1 mM MgCl$_2$, 0.2 mM CaCl$_2$, 5 mM KCl, 1 mg/ml fatty acid-free BSA) and centrifuge for 90 min at 100.000 g. The supernatant was recovered and incubated with 15 ul polyPIPsomes containing 5% PI3P (Echelon Y-P3003). Samples without liposomes were used to control for pelleting of protein aggregates. After 30 min of incubation at room temperature, liposomes were recovered by centrifugation at 100.000 g

for 90 min. The supernatant was recovered and the pellet was resuspended in 20 µl of LBB. 5 µl of the total mixture and 1 and 10 equivalents of the 100.000 g supernatants and pellets, respectively, were analyzed by immunoblot using a goat anti-GST antibody. The liposome pelleting assay was performed three times.

### Quantification, statistical analysis, and structure prediction

Quantifications were performed with the Fiji open-source platform (*Schindelin et al., 2012*). Statistical analysis was performed with GraphPad Prism. The D'Agostino-Pearson test was applied to data sets to assess normality. If the data followed a normal distribution or the result of the normality test was not significant, an unpaired two-tailed Student *t*-test was performed to assess significance. If the distribution was not normal, a two-tailed Mann-Whitney test was used. Results are expressed as mean ± SEM with respect to the number of cells (n) for a representative experiment. Prediction of IDRs was achieved with the IUPred2A software, which assigns each residue a IUPred score that is the probability of it being part of a IDR (*Meszaros et al., 2018*).

### Antibodies

Polyclonal sera against kazrin for immunoblotting were generated in rabbits using an N-terminal (amino acids 1–176) and a C-terminal (amino acids 161–327) fragment of kazrin C fused to GST. The following commercial antibodies were used in this study: anti-RAB11 (610656, AB_397983), anti-RAB4 (610888, AB_398205), anti-rabaptin-5 (610676, AB_398003), anti-GM130 (610822, AB_398141), anti-GGA2 (612612, AB_399892), anti-clathrin heavy chain (610499, AB_397865), anti-p150 Glued (610473, AB_397845), anti-α-adaptin (610501, AB_2313949), anti-γ-adaptin (610386, AB_397769), anti-N-cadherin (610920, AB_398236), anti-β-catenin (610153, AB_397554), anti-p120-catenin (610134, AB_397537), anti-desmoglein (610273, AB_397669), from BD Biosciences; anti-pericentrin (4448, AB_304461) and anti-EHD1 (109311, AB_10859459), anti kazrin C (74114, AB_10863615) and anti-VPS35 (57632, AB_946126), from Abcam; anti-kinesin-1 heavy chain (133184, AB_2132389) from Santa Cruz Biotechnology, anti-EEA1 (3288, AB_2096811) from Cell Signalling Technology; anti-tubulin (T-6557, AB_477584) from Merck; anti-GFP (632380, AB_10013427) from Takara Bio; anti-Dynein Heavy Chain (Ab23905, AB_2096669) from Abcam; anti-TOMM20 (WH0009804M1, AB_1843992). Peroxidase-conjugated anti-mouse (A2554, AB_258008), anti-goat (A4174, AB_258138) and rabbit (A0545, AB_257896) IgGs were from Merck. Alexa Fluor 568 anti-mouse IgG (A11037, AB_2534095), Alexa Fluor 568 anti-rabbit IgG (A11036, AB_10563566), and Alexa Fluor 647 anti-rabbit IgG (A21245, AB_2535813), from Thermo Fisher Scientific.

## Acknowledgements

M Martínez for lentivirus plasmids and J Roig for CRISPR plasmids (Institute fro Molecular Biology, Barcelona). M Robinson (Cambridge Institute for Medical Research, Cambridge) for sending the construct of the GST-γ-adaptin ear and M Mapelli (European Institute of Oncology IRCCS, Milan) for the GST-LIC1 and LIC2 constructs. T Surrey and C Mitchel for sharing the porcine-purified dynactin complex (Centre for Genomic Regulation, Barcelona). F Ruhnow, S Colombo, and S Speroni for dynein and dynactin purification and motility assays (Centre for Genomic Regulation, Barcelona). This work was financed by BFU2017-82959-P to MIG, BES-2012–053341 to AB, and BES-2015–071691 to IHP, from the Spanish government. The Andor Dragonfly 505 and the Zeiss LSM780 microscopes were funded by EQC2018-004541 EU FeDer and CSIC1501/18, respectively.

## Additional information

### Competing interests

María Isabel Geli: Reviewing editor, *eLife*. The other authors declare that no competing interests exist.

## Funding

| Funder | Grant reference number | Author |
|---|---|---|
| Agencia Estatal de Investigación | BFU2017-82959-P | María Isabel Geli |
| Agencia Estatal de Investigación | PID2020-120053GB-I00 | María Isabel Geli |
| Ministerio de Ciencia, Innovación y Universidades | EQC2018-004541 EU FeDer | Elena Rebollo |
| Consejo Superior de Investigaciones Científicas | CSIC1501/18 | Elena Rebollo |
| Ministerio de Ciencia, Innovación y Universidades | BES-2015-071691 BES-2012-053341 | Ines Hernandez-Perez Adrian Baumann |

The funders had no role in study design, data collection and interpretation, or the decision to submit the work for publication.

## Author contributions

Ines Hernandez-Perez, Conceptualization, Data curation, Formal analysis, Validation, Investigation, Methodology, Writing – review and editing; Javier Rubio, Conceptualization, Formal analysis, Investigation, Methodology; Adrian Baumann, Investigation; Henrique Girao, Conceptualization, Data curation, Formal analysis, Investigation; Miriam Ferrando, Conceptualization, Resources, Software, Supervision, Investigation, Methodology; Elena Rebollo, Conceptualization, Resources, Software, Supervision, Funding acquisition, Validation, Methodology, Writing – review and editing; Anna M Aragay, Conceptualization, Data curation, Formal analysis, Supervision, Funding acquisition, Validation, Investigation, Methodology, Writing – original draft, Project administration, Writing – review and editing; María Isabel Geli, Conceptualization, Data curation, Formal analysis, Supervision, Funding acquisition, Validation, Investigation, Writing – original draft, Project administration

## Author ORCIDs

Javier Rubio ⬡ http://orcid.org/0000-0003-0320-5100
María Isabel Geli ⬡ http://orcid.org/0000-0002-3452-6700

## Decision letter and Author response

Decision letter https://doi.org/10.7554/eLife.83793.sa1
Author response https://doi.org/10.7554/eLife.83793.sa2

## Additional files

### Supplementary files
• MDAR checklist

### Data availability

All data presented or analyzed in the manuscript has been loaded in Dryad DOI https://doi.org/10.5061/dryad.k6djh9w9q.

The following dataset was generated:

| Author(s) | Year | Dataset title | Dataset URL | Database and Identifier |
|---|---|---|---|---|
| Geli MI | 2023 | Data from: Kazrin promotes dynein/dynactin-dependent traffic from early to recycling endosomes | https://doi.org/10.5061/dryad.k6djh9w9q | Dryad Digital Repository, 10.5061/dryad.k6djh9w9q |

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
