## [Editor Report]

In their paper, Hernandez-Perez et al. perform a detailed and solid analysis of kazrin, a widely expressed protein that appears to be involved in many diverse cellular processes, but whose exact function is unknown. The authors employ mouse embryonic fibroblasts and biochemistry to investigate the function of Kazrin and determine that Kazrin promotes the dynein/dynactin-dependent transport of early endosomes. These valuable findings will be of interest to those in the field of intracellular transport.

---

## [Decision Letter]

**Decision letter after peer review:**

Thank you for submitting your article "Kazrin is an endosomal adaptor for dynein/dynactin" for consideration by *eLife*. Your article has been reviewed by 3 peer reviewers, and the evaluation has been overseen by a Reviewing Editor and Suzanne Pfeffer as the Senior Editor. The reviewers have opted to remain anonymous.

Essential revisions:

1) After discussion, the reviewers thought it was necessary for the authors to directly test the ability of Kazrin to be a dynein-dynactin adaptor. It was suggested that the authors perform the in vitro experiments outlined by Reviewer 1 to provide direct support for the title of this manuscript. If the authors are unable to provide these data, the reviewers thought the authors should soften their language throughout the manuscript, and change their title to a more apt description of their results.

2) The reviewers agreed that the language used throughout the manuscript was too bold based on the presented data. Please go through the suggestions of each reviewer to change/soften the language to ensure the descriptive language, claims and conclusions match the data being presented.

3) Finally, the reviewers offered suggestions about the addition of references throughout the paper to cite prior work and to put the current results within the context of the field. Please address these suggestions.

*Reviewer #2 (Recommendations for the authors):*

I am listing below a few suggestions that the authors might wish to consider to buttress the conclusion of their work as summarized in the title. Obviously, addressing each and every one of my suggestions would go far beyond the timeframe that is reasonable for a revised version, and therefore I leave to the authors' judgement the choice of how many of them are they willing to tackle.

A first place with room for substantial improvement is that the reference section, which is a bit outdated. Two, in my opinion, key reviews will surely help authors to address this shortcoming: Erika Holzbaur's (doi: 10.1242/jcs.227132) and Samara Reck-Peterson's (doi: 10.1038/s41580-018-0004-3). Of note, the authors often cite reviews rather than primary reports, which favors the interests of the journal(s) publishing their reviews but precludes fair recognition of the contributing scientists.

A point that is unclear to me is the relationship of the suggested kazrin pathway with the RAB11 FIP3 pathway, traditionally regarded as the pathway connecting sorting endosomes with the perinuclear recycling endosome (Bouchet et al., 2016; Eathiraj et al., 2006; Horgan et al., 2010; Vetter et al., 2015) (I wonder if authors have filmed kazrin with RAB11 and RAB5). Using dominant-negative RAB mutants could help to clarify the involvement of specific RABs in the process.

Bioinformatic analysis of kazrin is limited to a (convincing) local alignment shown in Figure 5. I think that a more detailed description of the domain organization of kazrin compared to well-established adaptors, together with extensive use of AlphaFold, would help to buttress the argued resemblance.

How do authors explain the inhibition of clathrin-dependent endocytosis by kazrin OE? Is it that kazrin hijacks a large proportion of the available pool of clathrin? How do you envisage the physiological role of the kazrin::γ-adaptin interaction?

On figure 5, coIP of p150 with GFP-kazrin is shown. It would be as important as reassuring to include more subunits of dynactin/dynein in these experiments. Have authors performed shotgun sequencing of GFP-trapped immunoprecipitates? Other hits?

It would also be reassuring that a genetic intervention in the dynein pathway (δ LIC, dynamitin overexpression, shRNA against LIC, the dynein heavy chain, etc prevents the arrival of kazrin to the RE).

I wonder why authors are acquiring z-stacks to follow the movement of kazrin endosomes. This implies that cells might be receiving an excessive amount of photodamaging light that will affect, in our experience, both run length, and speed.

It would be nice to have a longer exposure of the Ptd-inositides on 3F. Perhaps the authors could carry out liposome pull-downs?

The bar diagrams on the right panels of Figure 6 are unsuitable for presentation. I strongly suggest using one diagram per condition and to add the corresponding histograms. Please use colors as well.

Strictly speaking, the panels shown in Figure 6A are not kymographs, they are 2D projections of time stacks showing the trajectories of the vesicles. A kymograph is a graph depicting position over time. The pictures can be improved, could you please show more examples (as S-Figure)?

The number of movies is exceedingly large and often unnecessary. One Video per condition would be enough and localization of kazrin to endosome membranes could be illustrated with orthogonal views of MIPs, displayed in the main section. Any colocalization coefficient?

A plausible explanation for the kazrin-kinesin-1 association detected is that kin-1 transports dynein to the plus-ends of microtubules.

Figure 5, S1: I am afraid I can't see any movement (neither here nor in the Video)

(44) EMT: please introduce abbreviations as they appear in the main text, irrespective of whether they were spelled out in the abstract. (51): somehow regulates … meaning?. (79) accumulation? >> intracellular signal of …(171): could be demonstrated >> was detected; (172-174) I don't understand the rationale, please spell it out. (176): The strongest binding, you haven't measured affinity (185): polyK >>> poly-Lys; GFP-kazrin condensates: the word condensates evokes in my mind membraneless organelles. Wouldn't it be better to say accumulations?

Bouchet, J., M.W. McCaffrey, A. Graziani, and A. Alcover. 2016. The functional interplay of Rab11, FIP3 and Rho proteins on the endosomal recycling pathway controls cell shape and symmetry. Small GTPases. 7:1-6.

Eathiraj, S., A. Mishra, R. Prekeris, and D.G. Lambright. 2006. Structural basis for Rab11-mediated recruitment of FIP3 to recycling endosomes. J Mol Biol. 364:121-135.

Horgan, C.P., S.R. Hanscom, R.S. Jolly, C.E. Futter, and M.W. McCaffrey. 2010. Rab11-FIP3 links the Rab11 GTPase and cytoplasmic dynein to mediate transport to the endosomal-recycling compartment. J Cell Sci. 123:181-191.

Vetter, M., R. Stehle, C. Basquin, and E. Lorentzen. 2015. Structure of Rab11-FIP3-Rabin8 reveals simultaneous binding of FIP3 and Rabin8 effectors to Rab11. Nat Struct Mol Biol. 22:695-702.

*Reviewer #3 (Recommendations for the authors):*

1. For statistical tests in Figure 1C, Figure 1-S3, and elsewhere, "n" is not defined. Given the relatively large number (>25), it appears to represent the number of individual cells that were analyzed rather than independent biological replicates. As a result, the very small p values are misleading and may not be informative about reproducibility or inter-experiment variability. At a minimum, the authors should more clearly define "n" in the figure legend in order to explain how many experiments were performed and how many cells were analyzed per experiment.

2. It is not clear how it was determined that kazirin spots seen via microscopy are condensates. The authors should at minimum provide an acknowledgement that this is speculative.

3. The authors alternate between using red-green and magenta-green for showing 2 channels in microscopy experiments. For consistency and to help out color-blind readers, it would be best to use just the magenta-green option.

---

## [Author Response]

Reviewer #2 (Recommendations for the authors):I am listing below a few suggestions that the authors might wish to consider to buttress the conclusion of their work as summarized in the title. Obviously, addressing each and every one of my suggestions would go far beyond the timeframe that is reasonable for a revised version, and therefore I leave to the authors' judgement the choice of how many of them are they willing to tackle.A first place with room for substantial improvement is that the reference section, which is a bit outdated. Two, in my opinion, key reviews will surely help authors to address this shortcoming: Erika Holzbaur's (doi: 10.1242/jcs.227132) and Samara Reck-Peterson's (doi: 10.1038/s41580-018-0004-3). Of note, the authors often cite reviews rather than primary reports, which favors the interests of the journal(s) publishing their reviews but precludes fair recognition of the contributing scientists.

We thank the reviewer for the remark. We have changed the references to the original articles showing the data, whenever the list of articles was not excessively long.

A point that is unclear to me is the relationship of the suggested kazrin pathway with the RAB11 FIP3 pathway, traditionally regarded as the pathway connecting sorting endosomes with the perinuclear recycling endosome (Bouchet et al., 2016; Eathiraj et al., 2006; Horgan et al., 2010; Vetter et al., 2015) (I wonder if authors have filmed kazrin with RAB11 and RAB5). Using dominant-negative RAB mutants could help to clarify the involvement of specific RABs in the process.

We very actively searched for interactions between kazrin C and RAB5, RAB11 and RAB4. Given the phenotype installed upon kazrin depletion, its subcellular localization and its interaction with AP-1, we expected RAB4 to interact with kazrin, (Deneka et al., 2003; Nagelkerken et al., 2000; Perrin et al., 2013). We did find some evidence suggesting that endogenous kazrin could specifically interact with overexpressed YFP-RAB4 in Cos7 cells (Author response image 1), but unfortunately, we could not trust the data because the antibody used to presumably recognize endogenous kazrin still recognized some cellular structures in MEF kazKO. Also, the results in cells over-expressing mCherry-kazrin C and RAB4-YFP were not consistent and the cells look extremely vacuolated. Finally, we could never detect any specific interaction between kazrin and RAB4 in biochemical assays with purified components or in immunoprecipitations assays (Author response image 2).

**Author response image 1. sa2fig1:** Endogenous kazrin C might interact with overexpressed RAB4 in Cos7 cells. Fluorescence micrographs of Cos7 cells transiently expressing RAB4-YFP, RAB11YFP or RAB5-GFP, fixed and stained with a rabbit serum against kazrin C and a A568conjugated secondary antibody. The single red and green channels and the merge of the same field are shown. Bar = 10 µm.

**Author response image 2. sa2fig2:** Kazrin does not seem to interact with the RAB4, RAB5 or RAB11 GTPases in vitro. A. Immunoblot of GFP-TRAP immunoprecipitations from nondenaturing extracts from Cos7 cells transiently expressing GFP, GFP-kazrin C or N and Cterminal truncations of kazrin C (amino acids 1 to 174, and amino acids 161 to 327, respectively), probed for the RAB4 GTPase (upper panel) or stained with Ponceau red (lower panel) for detection of GFP or GFP-kazrin C constructs in the immunoprecipitates. B. Immunoblot of pull down assays of glutathione-Sepharose beads coated with increasing concentrations of GTP loaded RAB4, RAB5 and RAB11 fused to GST, incubated in the presence of non-denaturing protein extracts from Cos7 cells, The membranes were probed for endogenous kazrin and stained with Ponceau red to detect GST or the GST-RAB constructs.

Regarding the functional interplay between RAB11-FIP3 and kazrin, the data indicates that they play a role in the same pathway (endocytic recycling via the REs), but they do so in different steps, with RAB11-FIP3 possibly working downstream of kazrin. RAB11-FIP3 interacts with RAB11 and it is mostly localized to RAB11-labeled compartments, the TGN and the RE (Hales et al., 2001; Horgan et al., 2004). Instead, kazrin is mostly recruited to the EEs and not to REs, as shown by our cell fractionation experiments and by its capacity to specifically bind PI3P. On the other hand, even though depletion of either protein installs a scattered pattern of Tfn-loaded compartments ((Horgan et al., 2010) and our present work), depletion of RAB11-FIP3 causes the dispersal of RAB11 compartments without affecting the distribution of EEs (Horgan et al., 2007), whereas depletion of kazrin causes the accumulation of peripheral EEs without affecting the RAB11 distribution (our present work). These observations suggest that kazrin works in the retrograde transport of EEs or in the generation and/or motility of EE/RE transport intermediates that have not yet lost their EE identity. In agreement, we see that the scattered Tfn compartments accumulating in kazKO cells colocalize with EEA1 (Author response image 3). RAB11-FIP3 instead, seems to play a role in either receiving EE/RE transport intermediates that have already acquired RAB11, or in positioning the RE compartment at the pericentriollar region, which might be key for the docking and fusion of incoming transport intermediates. Based on the overwhelming data demonstrating the key role of RAB11-FIP3 in positioning the RE in different contexts (Bouchet et al., 2017; Fielding et al., 2005; Horgan et al., 2007; Schiel et al., 2012; Wilson et al., 2005), we would rather favor this last possibility, even though it might play different roles in association with different motors. For simplicity, we have not included all this considerations in the manuscript, but we now mention in the discussion that kazrin and RAB-FIP3 possibly work in the same endocytic recycling pathway, but at different stages.

We thank you the reviewer for pointing it out.

**Author response image 3. sa2fig3:** Colocalization of internalized TxRTfn and EEA1 in kazKO MEF. Merged confocal fluorescence microscopy of kazKO MEF loaded with TxR-Tfn (magenta) for 20 min, fixed an stained with anti-EEA1 and A641-conjugated secondary antibodies (Green). Scale bar, 10 µm.

The TxR and A641 individual channels and the merge of magnified fields are shown on the right.

Bioinformatic analysis of kazrin is limited to a (convincing) local alignment shown in Figure 5. I think that a more detailed description of the domain organization of kazrin compared to well-established adaptors, together with extensive use of AlphaFold, would help to buttress the argued resemblance.

We indeed made the interesting exercise to look at the α fold predictions of kazrin and different bona fide or candidate dynein/dynactin adaptors, but the outcome was not very informative, since most of them have a long coiled-coil N-terminal domain, which mediates the multiple interactions with the dynactin and dynein complexes, followed by a cargo binding region that is often depicted as an intrinsically disordered region. We show the α fold prediction for the mouse kazrin A, BICDR1 and RAB11-FIP3 in Author response image 4 as a matter of examples. Therefore, we thought that the sequence comparison would be more meaningful, especially because when we run a Clustal Omega alignment, the phylogenetic tree indicates that the similarity between kazrin and the other adaptors is comparable to the similarity among them (Author response image 4).

**Author response image 4. sa2fig4:** Similarities between kazrin and other bona fide or candidate dynein/dynactin adaptors. A. Α fold prediction models for the indicated proteins. B. Clustal omega phylogenetic tree for the human kazrin C and the indicated human bona fide or candidate dynein/dynactin adaptors.

If the reviewer feels that the data might be of interest, we could put it in the supplementary data. We have leaved it out for the moment because we think the alignment is more meaningful.

How do authors explain the inhibition of clathrin-dependent endocytosis by kazrin OE? Is it that kazrin hijacks a large proportion of the available pool of clathrin? How do you envisage the physiological role of the kazrin::γ-adaptin interaction?

We think that overexpression of kazrin leads to sequestration of some factors required for the uptake step of receptor-mediated endocytosis. It might possibly be clathrin, but alterations of the actin cytoskeleton seen upon overexpression of kazrin might also play a role (Sevilla et al., 2008). As for the interaction with AP-1, we envision that kazrin might directly link the endosomal subdomains containing this clathrin adaptor to the tubulin network.

On figure 5, coIP of p150 with GFP-kazrin is shown. It would be as important as reassuring to include more subunits of dynactin/dynein in these experiments. Have authors performed shotgun sequencing of GFP-trapped immunoprecipitates? Other hits?

We completely agree with the reviewer that showing that GFP-kazrin C can pull down not only dynactin but also dynein was very important in the context of proposing that kazrin promotes dynein/dynactin endosome motility. We now show in figure 5H that GFP-kazrin C but not GFP pulls downs the dynein heavy chain from non-denaturing extracts. The mass-spectrometry analysis of GFP-kazrin C pull downs is also a very good suggestion but we get at the moment very little material when we express GFP-kazrin C at physiological levels, so we plan to perform proximity labelling with an integrated kazrin C-BioID chimera in the near future. We did though some pull downs with GST-kazrin C from brain extracts and we identified, besides α and β tubulin, Kif5 a kinesin 1 that might link kazrin to the endosomal Rab4/Gadkin/AP-1 pathway (Schmidt et al., 2009), cenexin 1 and 2, centriole subdistal appendage proteins that might connect kazrin to the REs (Hehnly et al., 2012), and cytocentrin and RILP that might link kazrin to the Ral GTPases. We have not validated these results yet though.

It would also be reassuring that a genetic intervention in the dynein pathway (δ LIC, dynamitin overexpression, shRNA against LIC, the dynein heavy chain, etc prevents the arrival of kazrin to the RE).

We completely agree with the reviewer that showing the effects of inhibiting dynein activity on GFP-kazrin C localization and the distribution of EEs was important in the context of the manuscript. We now include an experiment in figure 5E and F showing that treatment of cells with the dynein inhibitor ciliobrevin prevents accumulation of GFP-kazrin C in the pericentrin foci and, concomitantly causes the dispersal of EEs towards the cell periphery, similar to kazrin depletion. We have also added another experiment in figure 5-S2, showing that a chimera of GFP-kazrin C fused to a mitochondria targeting domain can convey mitochondria to the pericentriolar region in Cos7 cells, suggesting that kazrin C can promote transport or clustering of cargo other that endosomes towards or at the microtubule minus ends, when fused to the adequate targeting signal.

I wonder why authors are acquiring z-stacks to follow the movement of kazrin endosomes. This implies that cells might be receiving an excessive amount of photodamaging light that will affect, in our experience, both run length, and speed.

We agree with the reviewer, but we needed to do Z-stacks not to lose tracks that move within the Z axes. In any case, the speeds and the length of the tracks are within the ranges of published data (Flores-Rodriguez et al., 2011; Loubery et al., 2008), so we assume that photo-damaging is not a major problem in our samples when filmed for 90 seconds under the conditions used, and that in any case, it would affect similarly all cell types under analysis.

It would be nice to have a longer exposure of the Ptd-inositides on 3F. Perhaps the authors could carry out liposome pull-downs?

We have now included a more clear exposure of the lipid strip overlay assay and we have added a liposome pelleting assay in figure 3G that shows that kazrin C interacts with liposomes containing PI3P.

The bar diagrams on the right panels of Figure 6 are unsuitable for presentation. I strongly suggest using one diagram per condition and to add the corresponding histograms. Please use colors as well.

We have tried several ways to represent these data and we found that a line plot with different colours was the clearest. I hope this new format is adequate for the reviewer.

Strictly speaking, the panels shown in Figure 6A are not kymographs, they are 2D projections of time stacks showing the trajectories of the vesicles. A kymograph is a graph depicting position over time. The pictures can be improved, could you please show more examples (as S-Figure)?

We have now added more examples of time projections used in these experiments in figure 6-S1 and S2. For clarity, we have coloured in blue the fairly straight, longer trajectories, as opposed to the more confined movements that appeared as round dots in the time projections (coloured in red). It is difficult to improve the images of the 2D projections because they need to be contrasted to evidence the trajectories. Instead, we have now chosen to included more videos illustrating the differences observed in cells expressing endogenous or GFP-kazrin C versus kazKO cells or kazKO cells expressing GFP or GFPkazrin C-Nt. Movies 8 and 11 show the endosome motility in representative WT and kazKO cells (Video 8) and kazKO cells expressing GFP, GFP-kazrin C or GFP-kazrin CNt (Video 11). Movies 9 and 10 show endosome motility in four magnified fields of different WT and kazKO cells, where longer and faster motility events can be observed when endogenous kazrin is expressed. Movies 12 to 14 show endosome motility in four magnified fields of different kazKO cells expressing, GFP-kazrin C (Video 12), GFP (Video 13) and GFP-kazrin C-Nt (Video 14). Longer and faster movements can be observed in the different insets of Video 12, as compared with movies 13 and 14. Finally, as suggested by the reviewer, we have reworded kymographs to time projections.

The number of movies is exceedingly large and often unnecessary. One Video per condition would be enough and localization of kazrin to endosome membranes could be illustrated with orthogonal views of MIPs, displayed in the main section. Any colocalization coefficient?

We have reduced the number of movies by combining the ones addressing the same point in one file. Thus, in movies 3 and 5, we show four 3D reconstructions of GFP-kazrin C or GFP-kazrin C-Nt foci in association with TxRTfn-loaded endosomes. We have removed the videos showing the reconstruction of TxRTfn endosomes in GFP expressing cells, because they were indeed not very informative. In Video 4, we now include the two 3D reconstructions showing association of EDH-protein enriched subdomains with GFP-kazrin C. We have also included orthogonal views of some of these videos in the corresponding figures, as suggested. Finally, we have attempted the analysis of the association between the GFP-kazrin C and GFP-kazrin C-Nt foci with TxR-Tfn-loaded endosomes. In our hand, it was impossible to do it automatically because the foci are very small, they do not strictly co-localize (Tfn is in the endosome lumen and kazrin is in the cytosol), and the cytosolic and nuclear backgrounds are high and different for the GFP-kazrin C and GFP-kazrin C-Nt construct. What we have done is to prepare more than twenty 3D reconstructions for each case and identify manually the TxRTfn foci (EE) closest to the GFP-kazrin C or GFPkazrin C-Nt foci, by measuring with the line Fiji tool. The distance between the centroid of those foci was then measured with the same tool in the Video frame showing the maximal separation between these foci. The data shows more association of GFP-kazrin C with endosomes as compared to GFP-kazrin C-Nt (Figure 4-S3). To further reinforce the conclusion that the C-terminus mediates binding of kazrin to cellular membranes, we have also repeated up to 5 times the fractionation experiment shown in figure 4A and we consistently show that a higher portion of GFP-kazrin C co-fraccionates with the microsomal fraction, as compared with GFP-kazrin C-Nt (Figure 4B).

A plausible explanation for the kazrin-kinesin-1 association detected is that kin-1 transports dynein to the plus-ends of microtubules.

We now introduce this possibility in the discussion.

Figure 5, S1: I am afraid I can't see any movement (neither here nor in the Video)

We have modified the figure and we have added arrows to make it more clear. We have also added a second example (movies 6 and 7 and Figure 5-S1).

44 EMT: please introduce abbreviations as they appear in the main text, irrespective of whether they were spelled out in the abstract.

We have done accordingly.

51: somehow regulates … meaning?.

Has been changed to “directly or indirectly”.

79: accumulation? >> intracellular signal of …

It has been changed accordingly.

171: could be demonstrated >> was detected;

It has been changed accordingly.

172-174: I don't understand the rationale, please spell it out.

“indicating that kazrin might preferentially bind” has been changed to “indicating that kazrin binds”.

176: The strongest binding, you haven't measured affinity.

The assertion has been removed.

185: polyK >>> poly-Lys;

It has been changed accordingly.

GFP-kazrin condensates: the word condensates evokes in my mind membraneless organelles. Wouldn't it be better to say accumulations?

As suggested by reviewer 1, we have changed it to foci.

Reviewer #3 (Recommendations for the authors):1. For statistical tests in Figure 1C, Figure 1-S3, and elsewhere, "n" is not defined. Given the relatively large number (>25), it appears to represent the number of individual cells that were analyzed rather than independent biological replicates. As a result, the very small p values are misleading and may not be informative about reproducibility or inter-experiment variability. At a minimum, the authors should more clearly define "n" in the figure legend in order to explain how many experiments were performed and how many cells were analyzed per experiment.

We have now added the information on the sample size in figure legends and we have extended the information on the replicas in Materials and methods. In summary, for endosomal transport assays (Tfn uptake, recycling, endosome motility…), the experiments were performed at least twice with at least two independently seeded plates, with analogous results. Data from the different replicas of one of the experiments were pulled for the graphs. The cell fractionation experiments were performed at least three times in MEF and also confirmed in other cell types. The biochemical experiments to demonstrate proteinprotein interactions were performed at least twice with the adequate controls from unspecific binding. All relevant interactions were tested with at least two different approaches (i. e. pull downs with purified components, pull downs with cell extracts, immunoprecipitations from cells expressing GFP-kazrin C, or immunoprecipitations with endogenous kazrin C).

2. It is not clear how it was determined that kazirin spots seen via microscopy are condensates. The authors should at minimum provide an acknowledgement that this is speculative.

We actually used the word condensate not to imply that the C-terminus of the protein generates membrane-less compartments or induces liquid-liquid-phase separation. However, since there was a consensus among the reviewers that this point was not clear, we have change “condensates” for “foci” in the new version of the manuscript.

3. The authors alternate between using red-green and magenta-green for showing 2 channels in microscopy experiments. For consistency and to help out color-blind readers, it would be best to use just the magenta-green option.

We have changed all figures to the green-magenta version, as requested.

References

Baumbach, J., Murthy, A., McClintock, M.A., Dix, C.I., Zalyte, R., Hoang, H.T., and Bullock, S.L. (2017). Lissencephaly-1 is a context-dependent regulator of the human dynein complex. *eLife* 6.

Bouchet, J., Del Rio-Iniguez, I., Vazquez-Chavez, E., Lasserre, R., Aguera-Gonzalez, S., Cuche, C., McCaffrey, M.W., Di Bartolo, V., and Alcover, A. (2017). Rab11-FIP3 Regulation of Lck Endosomal Traffic Controls TCR Signal Transduction. J Immunol 198, 2967-2978.

Deneka, M., Neeft, M., Popa, I., van Oort, M., Sprong, H., Oorschot, V., Klumperman, J., Schu, P., and van der Sluijs, P. (2003). Rabaptin-5alpha/rabaptin-4 serves as a linker between rab4 and γ(1)-adaptin in membrane recycling from endosomes. Embo J 22, 2645-2657.

Fielding, A.B., Schonteich, E., Matheson, J., Wilson, G., Yu, X., Hickson, G.R., Srivastava, S., Baldwin, S.A., Prekeris, R., and Gould, G.W. (2005). Rab11-FIP3 and FIP4 interact with Arf6 and the exocyst to control membrane traffic in cytokinesis. EMBO J 24, 33893399.

Flores-Rodriguez, N., Rogers, S.S., Kenwright, D.A., Waigh, T.A., Woodman, P.G., and Allan, V.J. (2011). Roles of dynein and dynactin in early endosome dynamics revealed using automated tracking and global analysis. PLoS One 6, e24479.

Granger, E., McNee, G., Allan, V., and Woodman, P. (2014). The role of the cytoskeleton and molecular motors in endosomal dynamics. Seminars in cell & developmental biology 31, 20-29.

Gutierrez, P.A., Ackermann, B.E., Vershinin, M., and McKenney, R.J. (2017). Differential effects of the dynein-regulatory factor Lissencephaly-1 on processive dynein-dynactin motility. J Biol Chem 292, 12245-12255.

Hales, C.M., Griner, R., Hobdy-Henderson, K.C., Dorn, M.C., Hardy, D., Kumar, R., Navarre, J., Chan, E.K., Lapierre, L.A., and Goldenring, J.R. (2001). Identification and characterization of a family of Rab11-interacting proteins. J Biol Chem 276, 39067-39075. Hehnly, H., Chen, C.T., Powers, C.M., Liu, H.L., and Doxsey, S. (2012). The centrosome regulates the Rab11- dependent recycling endosome pathway at appendages of the mother centriole. Curr Biol 22, 1944-1950.

Horgan, C.P., Hanscom, S.R., Jolly, R.S., Futter, C.E., and McCaffrey, M.W. (2010). Rab11-FIP3 links the Rab11 GTPase and cytoplasmic dynein to mediate transport to the endosomal-recycling compartment. J Cell Sci 123, 181-191.

Horgan, C.P., Oleksy, A., Zhdanov, A.V., Lall, P.Y., White, I.J., Khan, A.R., Futter, C.E., McCaffrey, J.G., and McCaffrey, M.W. (2007). Rab11-FIP3 is critical for the structural integrity of the endosomal recycling compartment. Traffic 8, 414-430.

Horgan, C.P., Walsh, M., Zurawski, T.H., and McCaffrey, M.W. (2004). Rab11-FIP3 localises to a Rab11-positive pericentrosomal compartment during interphase and to the cleavage furrow during cytokinesis. Biochem Biophys Res Commun 319, 83-94.

Jha, R., Roostalu, J., Cade, N.I., Trokter, M., and Surrey, T. (2017). Combinatorial regulation of the balance between dynein microtubule end accumulation and initiation of directed motility. EMBO J 36, 3387-3404.

Loubery, S., Wilhelm, C., Hurbain, I., Neveu, S., Louvard, D., and Coudrier, E. (2008). Different microtubule motors move early and late endocytic compartments. Traffic 9, 492509.

Nagelkerken, B., Van Anken, E., Van Raak, M., Gerez, L., Mohrmann, K., Van Uden, N., Holthuizen, J., Pelkmans, L., and Van Der Sluijs, P. (2000). Rabaptin4, a novel effector of the small GTPase rab4a, is recruited to perinuclear recycling vesicles. Biochem J 346 Pt 3, 593-601.

Olenick, M.A., and Holzbaur, E.L.F. (2019). Dynein activators and adaptors at a glance. J Cell Sci 132.

Perrin, L., Lacas-Gervais, S., Gilleron, J., Ceppo, F., Prodon, F., Benmerah, A., Tanti, J.F., and Cormont, M. (2013). Rab4b controls an early endosome sorting event by interacting with the γ-subunit of the clathrin adaptor complex 1. J Cell Sci 126, 4950-4962.

Schiel, J.A., Simon, G.C., Zaharris, C., Weisz, J., Castle, D., Wu, C.C., and Prekeris, R. (2012). FIP3-endosome-dependent formation of the secondary ingression mediates ESCRT-III recruitment during cytokinesis. Nat Cell Biol 14, 1068-1078.

Schmidt, M.R., Maritzen, T., Kukhtina, V., Higman, V.A., Doglio, L., Barak, N.N., Strauss, H., Oschkinat, H., Dotti, C.G., and Haucke, V. (2009). Regulation of endosomal membrane traffic by a Gadkin/AP-1/kinesin KIF5 complex. Proc Natl Acad Sci U S A 106, 15344-15349.

Sevilla, L.M., Nachat, R., Groot, K.R., and Watt, F.M. (2008). Kazrin regulates keratinocyte cytoskeletal networks, intercellular junctions and differentiation. J Cell Sci 121, 3561-3569.

Tang, N., and Marshall, W.F. (2012). Centrosome positioning in vertebrate development. J Cell Sci 125, 4951-4961.

Wilson, G.M., Fielding, A.B., Simon, G.C., Yu, X., Andrews, P.D., Hames, R.S., Frey, A.M., Peden, A.A., Gould, G.W., and Prekeris, R. (2005). The FIP3-Rab11 protein complex regulates recycling endosome targeting to the cleavage furrow during late cytokinesis. Mol Biol Cell 16, 849-860.

Yamada, M., Toba, S., Yoshida, Y., Haratani, K., Mori, D., Yano, Y., Mimori-Kiyosue, Y., Nakamura, T., Itoh, K., Fushiki, S., et al. (2008). LIS1 and NDEL1 coordinate the plusend-directed transport of cytoplasmic dynein. EMBO J 27, 2471-2483.